# Parameter Space Representation Learning on Mixed-type Data

## Abstract

A significant challenge in representation learning is to capture latent semantics in data mixing continuous, discrete, and even discretized observations (called mixed-type data), encountering issues like inconsistent discoveries and redundant modeling. Recently, Bayesian flow networks (BFNs) offer a unified strategy to represent such mixed-type data in the parameter space but cannot learn low-dimensional latent semantics since BFNs assume the size of parameters being the same as that of observations. This raises a new important question: *how to learn latent semantics in parameter spaces rather than in observation spaces of mixed-type data*? Accordingly, we propose a novel unified *parameter space representation learning* framework, ParamReL, which extracts progressive latent semantics in parameter spaces of mixed-type data. In ParamReL, a *self-encoder* learns latent semantics from intermediate parameters rather than observations. The learned semantics are then integrated into BFNs to efficiently learn unified representations of mixed-type data. Additionally, a *reverse-sampling procedure* can empower BFNs for tasks including input reconstruction and interpolation. Extensive experiments verify the effectiveness of ParamReL in learning parameter space representations for latent interpolation, disentanglement, time-varying conditional reconstruction, and conditional generation. The code is available at `https://anonymous.4open.science/r/ICLR25-F087/README.md`.

## 1 Introduction

This work explores a new important question: *How to learn latent semantics in parameter spaces rather than in observation spaces of mixed-type data comprising continuous, discrete, and even discretized observations*? We propose a novel unified *parameter space representation learning* framework that utilizes the parameter spaces rather than the observation spaces for mixed-type data.

Representation learning (Bengio et al., 2013) aims to discover low-dimensional latent semantics from high-dimensional observations, widely applied in areas including computer vision (Li et al., 2023; Zhao et al., 2023a; Dong et al., 2023), and data analytics (Tonekaboni et al., 2022; Oublal et al., 2024). While the main focus has been on continuous-valued data (Kim & Mnih, 2018; Chen et al., 2018; Meo et al., 2024), it is more challenging to uncover semantics in discrete (Austin et al., 2021; Chen et al., 2023) and even discretized (Van Den Oord et al., 2017; Razavi et al., 2019) data. However, existing efforts often encounter issues like inconsistent discoveries and redundant modeling (Zhou et al., 2023; Krishnan et al., 2018). Recently, Bayesian flow networks (BFNs) (Graves et al., 2023; Song et al., 2024; Xue et al., 2024) emerged as a promising deep generative model. BFNs use multiple steps similar to diffusion models (Ho et al., 2020; Song et al., 2021) to refine parameters of an output distribution for reconstructing observations. Accordingly, BFNs offer a unified strategy to handle mixed-type data while enabling fast sampling. However, they struggle to capture low-dimensional latent semantics, raising the above open question.

Correspondingly, we propose a novel unified *Param*eter space *Re*presentation *L*earning framework, ParamReL, which leverages the multi-step generative learning of BFNs for representation learning on mixed-type data. ParamReL tackles this by performing representation learning in the parameter space to extract high-level latent semantics. The key insight lies in progressively self-encoding the intermediate parameters of BFNs, generating low-dimensional latent semantics step by step. Specifically, ParamReL adopts an architecture similar to BFNs but with two significant innovations: (1)

a *self-encoder* encodes intermediate parameters into lower-dimensional latent semantics, capturing gradual semantic changes throughout the multi-step generation process; and (2) a *conditional decoder*, which conditions on latent semantics and intermediate parameters, and forms the parameters of an *output distribution* for simulating observations. Additionally, ParamReL involves *a reverse-sampling procedure* customized for tasks like image reconstruction and interpolation. Variational inference method is used in learning ParamReL, where mutual information is used to promote disentangled latent semantic learning, resulting in distinct and meaningful representations.

We evaluate ParamReL in learning meaningful high-level latent semantics from both discrete and continuous-valued observations on benchmark data. The sampling and reverse-sampling mechanisms of ParamReL successfully perform tasks such as latent interpolation, disentanglement, time-varying conditional reconstruction, and conditional generation. Notably, the self-encoder reveals progressive semantics throughout flow steps, enabling ParamReL to generate semantics with improved clarity, while maintaining high quality of sample generation.

## 2 UNDERSTANDING BAYESIAN FLOW NETWORKS - AN ALTERNATIVE VIEW

Bayesian Flow Networks (BFNs) (Graves et al., 2023; Song et al., 2024; Xue et al., 2024) serve as deep generative models with a primary objective to learn an output distribution for generating observations. The distribution's parameters are learned by a neural network, which takes the posterior parameters of observations of inputs. Here, we try to understand BFNs from an alternative parameter perspective since these (posterior) parameters play a key role in BFNs. BFNs involves concepts such as input distribution, sender distribution and receiver distribution, to introduce BFNs, making it less accessible to readers unfamiliar with BFNs. Interested readers may refer to Appendix A.1 and (Graves et al., 2023) for the original illustrations.

Figure 1 shows $T$ steps of training and sample generation in BFNs, similar to diffusion models (Ho et al., 2020; Song et al., 2021). To train BFNs, we minimize the divergence between the ground-truth data distribution and the evolving output distributions over $T$ steps. At each step $t \in \{T, \ldots, 1\}$, an intermediate (posterior) parameter $\boldsymbol{\theta}_t$ is first updated using a Bayesian update function $h(\cdot)$ as $\boldsymbol{\theta}_t = h(\boldsymbol{\theta}_{t+1}, \mathbf{x}_{t+1})$, where $\mathbf{x}_{t+1}$ is the observation at step $t+1$. $\boldsymbol{\theta}_t$ is then fed into a neural network $\psi(\cdot)$ to form the parameters of output distribution, i.e., a decoder $p_O(\mathbf{x}_t|\psi(\boldsymbol{\theta}_t))$, for model training. After training, these intermediate output distributions can be employed to simulate observations during the sample generation process, replacing the actual observations at each step $t$.

By working in the parameter space, BFNs can uniformly model continuous, discrete, and discretized observations. For example, BFNs can use the mean of Gaussian distributions as parameter $\boldsymbol{\theta}$ to model continuous data or use the event probabilities of categorical distributions as $\boldsymbol{\theta}$ to study discrete data (see detailed settings for distributions in Table 2). However, BFNs cannot produce meaningful

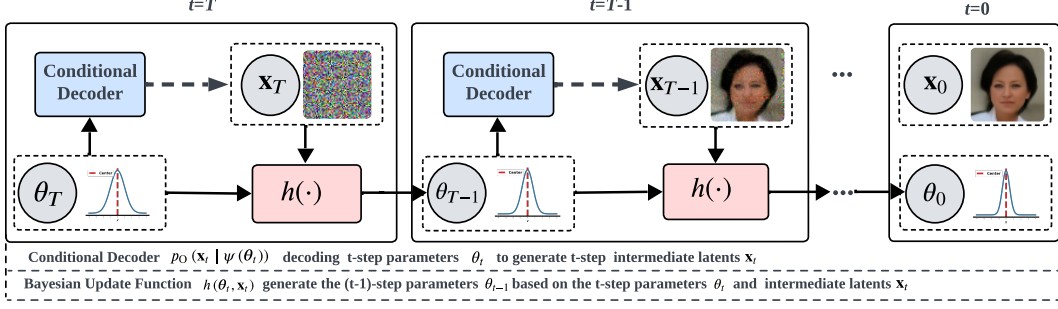

Figure 1: Our alternative understanding of BFNs. Each step consists of a conditional decoder $p_O(\mathbf{x}_t|\psi(\boldsymbol{\theta}_t)$ (in blue rectangle) and a Bayesian update function $h(\cdot)$ (in peach rectangle). In training BFNs, dashed arrows (between conditional decoder and $\{\mathbf{x}_t\}_{t=1}^T$) are non-existent as $\{\mathbf{x}_t\}_{t=1}^T$ refers to observations. The dashed arrows become solid for sample generation, representing the decoder generates $\mathbf{x}_t$ in sample generation.

latent semantics capturing high-level concepts in the mixed-type observations, such as hair colors in portrait images.

## 3 ParamReL: Parameter Space Representation Learning

Here, we explain the framework of ParamReL and its main design mechanisms.

### 3.1 The ParamReL Framework

The framework and workflow of ParamReL are shown in Figure 2. ParamReL leverages the parameter space for representation learning by extracting low-dimensional latent semantics from high-dimensional mixed-type data. Different from BFNs in approximating data distribution $p(\mathbf{x}_0)$, ParamReL learns the joint distribution over observation $\mathbf{x}_0$ and a series of latent semantics $\{\mathbf{z}_t\}_{t=1}^T$, with $|\mathbf{z}_t| \ll |\mathbf{x}_0|, \forall t \in \{1, \ldots, T\}$. That is, ParamReL seeks to reconstruct $\mathbf{x}_0$ while obtaining meaningful low-dimensional latent semantics $\{\mathbf{z}_t\}_{t=1}^T$.

Building on BFNs, ParamReL consists of four main components:

(1) *A self-encoder*, conditioning on the intermediate (posterior) parameters $\boldsymbol{\theta}_t$ to generate progressive latent semantics $\mathbf{z}_t$, described in Section 3.2.

(2) *A conditional decoder*, using a neural network on latent semantics $\mathbf{z}_t$ and intermediate parameters $\boldsymbol{\theta}_t$ to form the output distribution for subsequent steps, detailed in Section 3.3.

(3) *A sampling and reverse-sampling process*, facilitating tasks such as image reconstruction and interpolation, outlined in Section 3.4.

(4) *A training and testing procedure*, as discussed in Section 3.5, optimizing latent semantics $\mathbf{z}_t$ and ensuring effective model generalization.

Together, ParamReL forms a robust framework to capture and utilize latent semantics and to improve the performance of tasks including unconditional image generation and reconstruction.

### 3.2 Parameter Encoding through A Self-encoder

The *self-encoder*, denoted as $q_\phi(\mathbf{z}_t|\boldsymbol{\theta}_t, t)$, progressively encodes intermediate parameters $\boldsymbol{\theta}_t$ into low-dimensional latent semantics $\mathbf{z}_t$, which facilitates representation learning from high-dimensional, mixed-type data at each step $t$. (Baranchuk et al., 2021) has shown that upsampling

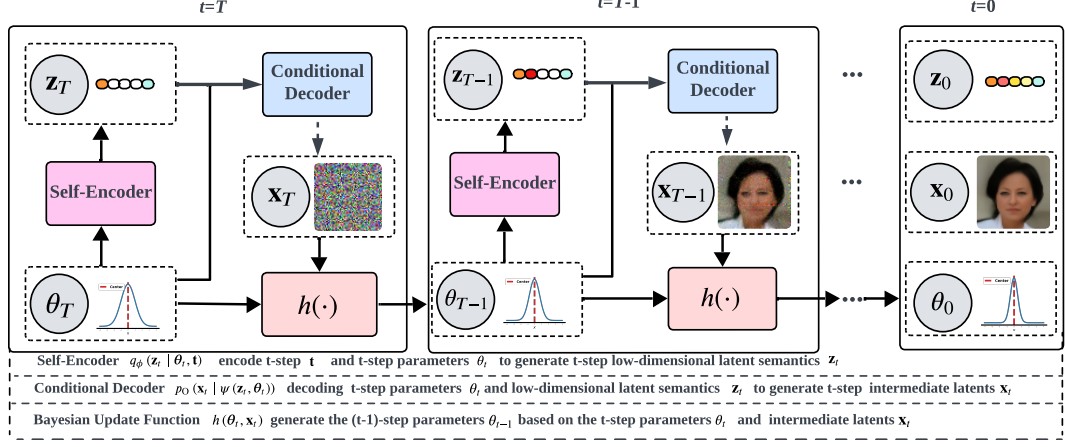

Figure 2: The framework of ParamReL. Each step consists of a self-encoder $q_\phi(\mathbf{z}_t|\boldsymbol{\theta}_t, t)$ (pink rectangle), a conditional decoder $p_O(\mathbf{x}_t|\psi(\mathbf{z}_t, \boldsymbol{\theta}_t))$ (blue rectangle), and Bayesian update $h(\cdot)$ (peach rectangle). During the reverse-sampling stage, the self-encoder $q_\phi$ encodes intermediate parameters $\boldsymbol{\theta}_t$ into a time-specific latent semantic $\mathbf{z}_t$, and $p_O(\mathbf{x}_t|\psi(\mathbf{z}_t, \boldsymbol{\theta}_t))$ generates $\mathbf{x}_t$.

layers from a U-Net in pre-trained diffusion models (Rombach et al., 2022) may capture meaningful semantic information. Inspiring from this discovery and in training ParamReL, we adopt approaches similar to (Luo et al., 2024) to parameterize $q_\phi(\mathbf{z}_t|\boldsymbol{\theta}_t, t)$ (see Appendix C.1 for more details). Through $q_\phi(\mathbf{z}_t|\boldsymbol{\theta}_t, t)$, the intermediate parameter $\boldsymbol{\theta}_t$ effectively encodes itself into $\mathbf{z}_t$, together they form $\psi(\boldsymbol{\theta}_t, \mathbf{z}_t)$ for the output distribution.

Ideally, the latent semantics $\mathbf{z}_t$ should provide low-dimensional semantics distinct from the intermediate parameters $\boldsymbol{\theta}_t$ in BFNs but without compromising the data reconstruction process. To learn high-quality latent semantics, a smooth, learnable latent space is necessary, which is ensured by integrating the prior distribution $p(\mathbf{z}_t)$ into a robust probabilistic framework, allowing efficient sampling of $\mathbf{x}_0$. For simplicity and efficiency, we assume $p(\mathbf{z}_t)$ follows a Gaussian distribution.

$q_\phi(\mathbf{z}_t|\boldsymbol{\theta}_t, t)$ differs from traditional auto-encoders $q_\phi(\mathbf{z}|\mathbf{x}_0)$ in two key aspects:

- $q_\phi(\mathbf{z}_t|\boldsymbol{\theta}_t, t)$ is conditioned on the intermediate parameter $\boldsymbol{\theta}_t$, rather than being conditioned on $\mathbf{x}_0$. This summarizes information from all previous steps to enable generating latent semantic $\mathbf{z}_t$ through all the $T$ steps.

- The self-encoder generates a step-wise semantic $\mathbf{z}_t$, which is tailored to the dynamic behavior of variables over time $t$. This series of latent semantics $\{\mathbf{z}_t\}_{t=1}^T$ are expected to exhibit progressive semantic behaviors (such as gradual changes in age, smile, or skin color) throughout the generation process (as illustrated in the right panel of Figure 13).

When observations $\mathbf{x}_0$ are unavailable, e.g. sample generation tasks, it is also worth noting that directly using regular auto-encoders like $q_\phi(\mathbf{z}|\mathbf{x}_0)$ to generate latent semantics is infeasible. They may require an additional module to generate latent semantics (Preechakul et al., 2022), while training such modules would introduce computational overhead. However, in their case, not using auto-encoders $q_\phi(\mathbf{z}|\mathbf{x}_0)$ would lead to inefficient resource use.

### 3.3 CONDITIONAL DECODER

The conditional decoder refers to the output distribution $p_O(\mathbf{x}_t|\psi(\boldsymbol{\theta}_t, \mathbf{z}_t))$ which conditions on latent semantics $\mathbf{z}_t$ and intermediate parameter $\boldsymbol{\theta}_t$ to simulate $\mathbf{x}_t$. The condition $\psi(\boldsymbol{\theta}_t, \mathbf{z}_t)$ explicitly incorporates $\mathbf{z}_t$ as part of its conditioning mechanism. Following the settings in diffusion models (Ho et al., 2020; Song et al., 2021), we use the U-Net architecture with the Cross-Attention in each layer specified as

$$\text{Cross-Attention}(\boldsymbol{\theta}_t, \mathbf{z}_t) = \text{softmax}(\frac{\mathbf{Q}\mathbf{K}^\top}{\sqrt{d}})\mathbf{V}, \text{ where } \mathbf{Q} = \mathbf{W}^Q\boldsymbol{\theta}_t, \mathbf{K} = \mathbf{W}^K\mathbf{z}_t, \mathbf{V} = \mathbf{W}^V\mathbf{z}_t$$

where $\mathbf{W}^Q, \mathbf{W}^K, \mathbf{W}^V$ are the query, key and value weight matrix, respectively. See the detailed U-Net architecture in Appendix C.2.

Since $\mathbf{z}_t$ works together with the corresponding intermediate parameter $\boldsymbol{\theta}_t$, it is expected that $\mathbf{z}_t$ aligns well with the progressively structured parameter $\boldsymbol{\theta}_t$. Lower-level intermediate latent $\mathbf{x}_t$ (such as hair texture) is progressively incorporated. The proposed self-encoder works consistently with the conditional decoder here as both work on $\boldsymbol{\theta}_t$, see Figure 6 (b).

### 3.4 SAMPLING AND REVERSE-SAMPLING PROCESSES

After training ParamReL, the sampling and reverse-sampling processes play a crucial role in generating and reconstructing data, which is essential for tasks such as image generation and interpolation. Generating samples begins with an initial guess of the intermediate parameters $\boldsymbol{\theta}_{T+1}$. From $\boldsymbol{\theta}_{T+1}$, this sampling process sequentially generates $\mathbf{x}_T, \mathbf{x}_{T-1}, \dots, \mathbf{x}_0$. Specifically, given the parameter $\boldsymbol{\theta}_t$ at each step $t$, we have:

$$\mathbf{z}_t \sim q_\phi(\mathbf{z}_t|\boldsymbol{\theta}_t, t), \ \mathbf{x}_t \sim p_O(\mathbf{x}_t|\psi(\boldsymbol{\theta}_t, \mathbf{z}_t)), \ \boldsymbol{\theta}_{t-1} = h(\boldsymbol{\theta}_t, \mathbf{x}_t). \tag{1}$$

We use the trained encoder $q_\phi(\mathbf{z}_t|\boldsymbol{\theta}_t, t)$ to replace the prior $p(\mathbf{z}_t)$ of $\mathbf{z}_t$ for improving the sampling quality. After $\boldsymbol{\theta}_0$ is obtained, a sample can be generated as $\mathbf{z}_0 \sim q_\phi(\mathbf{z}_0|\boldsymbol{\theta}_0, 0), \mathbf{x}_0 \sim p_O(\mathbf{x}_0|\psi(\boldsymbol{\theta}_0, \mathbf{z}_0))$.

However, the reverse-sampling process, which transits the observation $\mathbf{x}_0$ through the intermediate latents $\mathbf{x}_1, \mathbf{x}_2, \dots, \mathbf{x}_{T-1}$ until $\mathbf{x}_T$, is not as straightforward as the sampling procedure. Without a

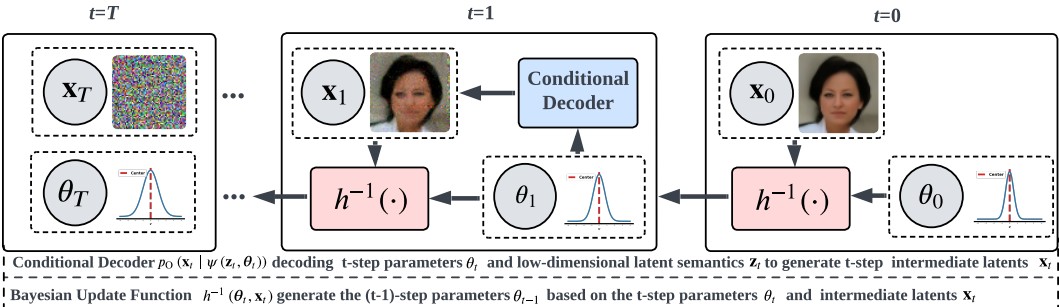

Figure 3: Reverse-sampling process in BFNs.

clearly defined reverse-sampling process, it would be challenging to perform tasks such as image reconstruction and interpolation. In fact, by taking the inverse of the Bayesian update function $h(\cdot)$ as $\boldsymbol{\theta}_t = h^{-1}(\boldsymbol{\theta}_{t-1}, \mathbf{x}_{t-1})$, the intermediate latent $\mathbf{x}_{t-1}$ can transit to $\mathbf{x}_t$ as:

$$\boldsymbol{\theta}_t = h^{-1}(\boldsymbol{\theta}_{t-1}, \mathbf{x}_{t-1}), \ \mathbf{z}_t \sim q_{\boldsymbol{\phi}}(\mathbf{z}_t | \boldsymbol{\theta}_t, t), \ \mathbf{x}_t \sim p_O(\mathbf{x}_t | \psi(\boldsymbol{\theta}_t, \mathbf{z}_t)). \quad (2)$$

Given the straightforward definition of Bayesian update function $h(\cdot)$, its inverse operation is generally easy to derive. The details of such results can be found in Figure 14. Furthermore, this developed reverse-sampling process can be naturally extended to BFNs. Transiting $\mathbf{x}_{t-1}$ to $\mathbf{x}_t$ at time $t$ can be performed as $\boldsymbol{\theta}_t = h^{-1}(\boldsymbol{\theta}_{t-1}, \mathbf{x}_{t-1})$, with $\mathbf{x}_t$ sampled as $\mathbf{x}_t \sim p_O(\mathbf{x}_t | \psi(\boldsymbol{\theta}_t))$. With this approach, BFNs can effectively perform tasks like image reconstruction and interpolation, which were difficult or even impossible by previous BFNs models. Figure 3 shows the reverse-sampling process of BFNs. The ParamReL version is provided in Figure 7 in Appendix A.

### 3.5 TRAINING AND TEST WITH PARAMREL

Here, we outline the process of training and testing ParamReL by focusing on optimizing ParamReL to learn meaningful latent semantics while ensuring effective reconstruction of observations. The training process involves variational inference to approximate the joint distribution of latent variables, and a mutual information term is integrated into improving the quality of learned latent semantics by strengthening the relationship between intermediate parameters and latent semantics.

**Variational Inference for Intractable Joint Distribution** In ParamReL, the joint distribution over $\mathbf{x}_0$, intermediate latents $\{\mathbf{x}_t\}_{t=1}^T$ and latent semantics $\{\mathbf{z}_t\}_{t=1}^T$ can be defined as $p(\mathbf{x}_0, \{\mathbf{x}_t\}_{t=1}^T, \{\mathbf{z}_t\}_{t=1}^T | -) = p_O(\mathbf{x}_0 | \psi(\boldsymbol{\theta}_0, \mathbf{z}_0)) \cdot \prod_{t=1}^T [p(\mathbf{z}_t) \mathbb{E}_{p_O(\mathbf{x}_t | \psi(\boldsymbol{\theta}_t, \mathbf{z}_t))}[p_S(\mathbf{x}_{t-1} | \mathbf{x}_t)]]$, where the output distribution $p_O(\mathbf{x}_0 | \psi(\boldsymbol{\theta}_0, \mathbf{z}_0))$ at step 0 is used to model observation $\mathbf{x}_0$, and $\mathbb{E}_{p_O(\mathbf{x}_t | \psi(\boldsymbol{\theta}_t, \mathbf{z}_t))}[p_S(\mathbf{x}_{t-1} | \mathbf{x}_t)]$ follows the definition of BFNs to model intermediate latent $\mathbf{x}_{t-1}$, and $p_S(\mathbf{x}_{t-1} | \mathbf{x}_t)$ is a noisy distribution of $\mathbf{x}_t$.

With $q_{\boldsymbol{\phi}}(\mathbf{z}_t | \boldsymbol{\theta}_t, t)$ defined as the encoder for $\mathbf{z}_t$ and $p_S(\mathbf{x}_{t-1} | \mathbf{x}_t)$ defined as the variational distribution for $\mathbf{x}_{t-1}$, the evidence lower bound (ELBO) on the marginal log-likelihood of observation $\mathbf{x}_0$ is (see the full derivation in Appendix B):

$$\log p(\mathbf{x}_0) \geq -\sum_{t=1}^T \mathbb{E}_{p_F(\boldsymbol{\theta}_t | -)} \mathbb{E}_{q_{\boldsymbol{\phi}}(\mathbf{z}_t | \boldsymbol{\theta}_t, t)} \left\{ \text{KL} \left[ p_S(\mathbf{x}_{t-1} | \mathbf{x}_0) \ \| \ \mathbb{E}_{p_O(\mathbf{x}_t | \psi(\boldsymbol{\theta}_t, \mathbf{z}_t))}[p_S(\mathbf{x}_{t-1} | \mathbf{x}_t)] \right] \right.$$

$$\left. -\text{KL} \left[ q_{\boldsymbol{\phi}}(\mathbf{z}_t | \boldsymbol{\theta}_t, t) \ \| \ p(\mathbf{z}_t) \right] \right\} + \mathbb{E}_{p_F(\boldsymbol{\theta}_0 | -) q_{\boldsymbol{\phi}}(\mathbf{z}_0 | \boldsymbol{\theta}_0, 0)} \left[ \ln p_O(\mathbf{x}_0 | \psi(\boldsymbol{\theta}_0, \mathbf{z}_0)) \right] := \text{ELBO}. \quad (3)$$

Maximizing ELBO is equivalent to performing amortized inference (Kingma & Welling, 2014) through encoders $q_{\boldsymbol{\phi}}(\mathbf{z}_t | \boldsymbol{\theta}_t, t)$ and learning likelihood function through decoders (Zhao et al., 2019). When the encodable posterior $q_{\boldsymbol{\phi}}(\mathbf{z}_t | \boldsymbol{\theta}_t, t)$ is used to infer high-level semantics $\mathbf{z}_t$, those intermediate latents $\{\mathbf{x}_t\}_{t=1}^T$ contain low-level information in generating the observations. In ParamReL, the parameters of the output distribution are learned through iteratively proceeding the Bayesian updating functions and a learned noise model $\psi(\boldsymbol{\theta}, \mathbf{z})$ parameterized by neural networks $\psi$.

**Mutual Information Regularization** Ideally, during the training phase, we want to acquire the latent semantic $\mathbf{z}_t$ by the self-encoder $q_\phi(\mathbf{z}_t|\boldsymbol{\theta}_t, t)$ and achieve high-quality reconstruction $\widehat{\mathbf{x}_0}$ by the decoder (i.e., the output distribution $p_O(\mathbf{x}_0|\psi(\boldsymbol{\theta}_0, \mathbf{z}_0))$). However, there exists a trade-off between inference and learning (Shao et al., 2020; Wu et al., 2024) coherent in optimizing the ELBO in Eq. (3). In most cases, optimizing ELBO favours fitting likelihood rather than inference (Zhao et al., 2019). Based on the rate-distortion theory (Alemi et al., 2018; Bae et al., 2023), the rate, represented by the KL divergence term constrained by the encoders, compresses sufficient information to minimize the distortion, or reconstruction error, while simultaneously limiting the informativeness to promote a smooth latent space.

To remedy the insufficient representation learning during the inference stage, we want to increase the dependence between intermediate parameters $\boldsymbol{\theta}_t$ and latent semantics $\mathbf{z}_t$ by maximizing their mutual information $I(\boldsymbol{\theta}_t, \mathbf{z}_t)$. We can rewrite the tractable learning object in ParamReL by adding the mutual information maximization term as $\text{ELBO}_+ = \text{ELBO} + \frac{\gamma}{T}\sum_t I_q(\boldsymbol{\theta}_t; \mathbf{z}_t)$, where $\gamma$ is the trade-off parameter. Considering that we cannot optimize this object directly, we can rewrite it by factorizing the rate term into mutual information and total correlation (TC), see details in Appendix B.

## 4 RELATED WORK

Recent advances have demonstrated that diffusion models (Ho et al., 2020; Song et al., 2021) are capable of generating high-quality data. Nonetheless, compared to the autoencoder framework, the intermediate outputs in diffusion stages are high-dimensional and lack smoothness, making them unsuitable for representation learning. Contemporary research focuses on encoding a conditional latent space to acquire low-dimensional semantic representations. However, those observations-based models (Preechakul et al., 2022; Wang et al., 2023), such as VAEs and diffusion models, exhibit limitations when applied to discrete data.

Deep hierarchical VAEs have seen progress in capturing latent dependence structures for encoding an expressive posterior, statistically or semantically. VQVAE-based (Van Den Oord et al., 2017; Razavi et al., 2019) models have local-to-global features-based explanatory hierarchies at the image level, forming a codebook-based discrete posterior. In (Sønderby et al., 2016; Tomczak & Welling, 2018), recursive latent structures in multi-layer networks form an aggregated posterior. NVAE (Vahdat & Kautz, 2020) demonstrates that depth-wise hierarchies encoded by residual networks can approximate the posterior precisely despite using shallow networks. Unlike the observation-based encoder, where the information flow between input and latent is maximized in encoding-decoding pipelines in the sample space, ParamReL uses progressive encoders in the parameter space to capture the dynamic semantics.

Pre-trained diffusion models (Rombach et al., 2022), (Baranchuk et al., 2021) have shown that the upsampling features from a U-Net can capture semantic information useful for downstream tasks. This discovery has sparked increasing research in leveraging these upsampling features of pre-trained diffusion models across various applications, including classification (Xiang et al., 2023; Mukhopadhyay et al., 2023), semantic segmentation (Baranchuk et al., 2021; Zhao et al., 2023b), panoptic segmentation (Xu et al., 2023), semantic correspondence (Tang et al., 2023; Zhang et al., 2024; Luo et al., 2024; Hedlin et al., 2024), and image editing (Tumanyan et al., 2023; Hertz et al., 2022). In most of these approaches, identifying the optimal denoising step and upsampling layer is crucial for achieving high predictive performance. These approaches do not suggest fundamental changes to model architectures or training methodologies, leaving the specific architectural components and techniques for learning useful semantic representations unclear. ParamReL uses these discoveries to construct efficient self-encoders.

## 5 EXPERIMENTS

We present two ParamReL variants operating in different parameter spaces: ParamReLd for discrete input distributions (Section 5.2), and ParamReLc for continuous input distributions (Section 5.3), respectively. We evaluate the representation learning capabilities of ParamReL in three reconstruction-based tasks: latent interpolation, disentanglement, and time-varying conditional reconstruction. Additionally, we evaluate the model for unconditional generation, where samples are generated *only from the decoder* using a given prior.

## 5.1 Evaluation Setup

We conduct a two-fold comparison to evaluate the performance of ParamReL variants. Firstly, we compare our parameter-based models (ParamReLc and ParamReLd) with established sample-based representation learning baselines, including AE and VAE-based models such as $\beta$-VAE (Higgins et al., 2017), infoVAE (Zhao et al., 2019), and diffusion-based models such as DiffAE (Preechakul et al., 2022) and InfoDiffusion (Wang et al., 2023). These models represent key advancements in the field: $\beta$-VAE introduce disentanglement into VAE, infoVAE incorporates MMD for balancing generation and representation, while DiffAE and InfoDiffusion explore the integration of AEs and VAEs into diffusion models to learn encodable latents and disentangled representations, respectively. Secondly, we compare the performance of ParamReLc and ParamReLd across various input distributions for continuous and discrete data, respectively. The discrete datasets include binarized versions of MNIST (bMNIST) (Deng, 2012), FashionMNIST (bFashionMNIST) (Xiao et al., 2017), while the continuous datasets include CelebA (Liu et al., 2015), CIFAR10 (Krizhevsky & Hinton, 2009), and Shapes3D (Burgess & Kim, 2018)[1]. The detailed hyperparameter choices and experimental configurations for each dataset are provided in Appendix C.3. This comparison allows for a detailed examination of how different parameter space assumptions impact the representation learning of discrete and continuous data.

## 5.2 Semantic Representation of Discrete Data by ParamReLd

Here, we measure the quality of the learned latent semantics $\mathbf{z}_0$ through the downstream classification tasks. Since $\mathbf{z}_0$ locates at step 0, they should be *general* and *transferable* (Franceschi et al., 2019). Various datasets by deep classifiers are assessed to ensure their universality. Specifically, following the approach in Xiao & Bamler (2023), we train a classifier on labeled test sets for each ParamReL model. We allocate $80\%$ of the dataset for training a classifier and reserve the remaining $20\%$ for test purposes. The performance on the test set is evaluated based on AUROC. This process is conducted in a 5-fold cross-validation manner, with the results reported as mean $\pm$ one standard deviation. The results are shown in Figure 4 (a). Higher AUROC suggests that the learned latent semantics $\mathbf{z}_0$ contain more information about data. In addition to assessing the representation quality, we also compare the image reconstruction ability against baselines. From the FID values in Figure 4 (a) and Figures 11, 12 in Appendix E.3, we can conclude that VAE-based models still produce blurry reconstructions, while diffusion-based and parameter-based models can build near-exact reconstructions. Refer to Figure 11 and Figure 12 in Appendix E.3 for the generated binary images.

## 5.3 Semantic Representation of Continuous Data by ParamReLc

On continuous data, we evaluate ParamReLc for conditional generation, conditional reconstruction, latent interpolation, and disentanglement.

**High-level Representation Learning for Conditional Generation** Figure 13 (a) in Appendix E.2 demonstrates that high-level semantic information is captured by the learned latent semantics $\{\mathbf{z}_t\}_{t=1}^T$ for image generation. This is illustrated by a set of latent-sample pairs $< \{\mathbf{z}_t^i\}_{t=1}^T, \mathbf{x}_T^{i,j} >$, where $\{\mathbf{z}_t^i\}_{t=1}^T$ are obtained by reverse-sampling from the $i$-th input image through the trained ParamReL, and $\mathbf{x}_T^{i,j}$ is the $j$-th sample from $\mathcal{N}(\mathbf{0}, \mathbf{I})$ corresponding to the $i$-th input image. Concurrently, the low-level information, such as local attributes in images (e.g., Narrow_Eyes, Mouth_Slightly_Open, Blond_Hair), are determined by $\mathbf{x}_T^{i,j}$.

**Time-varying Representation Learning for Conditional Reconstruction** We design a *new* time-varying reconstruction task to evaluate the effectiveness of the progressive latent semantics learned by the self-encoder. A latent-sample pair $< \{\mathbf{z}_t^{\text{fixed}}\}_{t=1}^T, \mathbf{x}_T^{\text{fixed}} >$ is first obtained by apply the trained ParamReL's reverse-sampling process on an image. Then, we use the latent semantics at step $t^*$ to replace other steps' ones and "reconstruct" the image as $\mathbf{x}_t \sim p_O(\mathbf{x}_t|\psi(\boldsymbol{\theta}_t, \mathbf{z}_{t^*}^{\text{fixed}})), \boldsymbol{\theta}_{t-1} = h(\boldsymbol{\theta}_t, \mathbf{x}_t), \forall t = T, \ldots, 1$. In that case, the attributes vary due to the semantics evolution encoded by time-specific latent. Refer to Figure 4 (b), and Figure 13 (b) in Appendix E.2 for more explanation.

---

[1]For the discrete version, continuous data ($k$-bit images) can be discretized into $2^k$ bins by dividing the data range $[-1, 1]$ into $k$ intervals, each of length $2/k$.

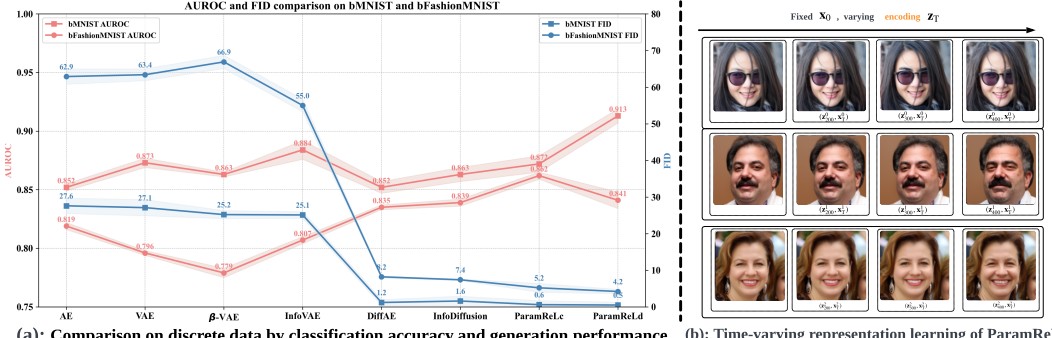

(a): Comparison on discrete data by classification accuracy and generation performance.    (b): Time-varying representation learning of ParamReL

Figure 4: Quantitative representation learning comparison over generative models on discrete data (a). ParamReL demonstrates competitive performance in capturing semantic information for classification, achieving approximately $0.84$ AUROC for bFashionMNIST and $0.91$ for bMNIST. Additionally, it shows robust generative capabilities, with FID values ranging from $0.5$ to $0.6$ for bMNIST and around $5$ for bFashionMNIST. Among the ParamReL-based models, ParamReLd with a categorical distribution is particularly effective in modelling discrete data distributions, yielding lower FID values of $0.5$ for bMNIST and $4.2$ for bFashionMNIST. As shown in (b), the learned semantics exhibit progressive, time-varying changes. By varying time encodes at 200, 300, 400 time steps, more attributes will be influenced in the reconstruction stage: the Wavy_hair, Brown_hair, Arched_Eyebrows attributes in the first line, the Double_Chin, Mustache, Goatee attributes in the second line and the Young, High_Cheekbones, Arched_Eyebrows attributes in the third line. Notations: [AUROC, FID]; [(●, bMNIST), (■, bFashionMNIST)]; [(−, ParamReLd),(− · −, ParamReLc)].

Table 1: Comparison of representation learning algorithms on continuous data by disentanglement performance (mean ± std) and classification. The quantitative results for each algorithm are averaged over five trials. Notations: Modeling on data space $\mathcal{D}$, parameter space $\mathcal{P}$. Prior distributions: Gaussian $g$, Categorical $c$, Delta $d$. ↑: higher better, ↓: lower better. Color: **Top-1**, Top-2.

| Prior on | Prior type | Methods | CelebA | | | | Shapes3D | | CIFAR-10 | |
|---|---|---|---|---|---|---|---|---|---|---|
| | | | $\mathcal{TAD}$ ↑ | $\mathcal{ATTRS}$ ↑ | $\mathcal{FID}$ ↓ | $\mathcal{AUROC}$ ↑ | $\mathcal{DCI}$ ↑ | $\mathcal{AUROC}$ ↑ | $\mathcal{FID}$ ↓ | $\mathcal{AUROC}$ ↑ |
| $\mathcal{D}$ | - | AE | 0.042 ±0.004 | 1.0 ±0.0 | 90.4±1.8 | 0.759 ±0.003 | 0.219 ±0.001 | 0.796±0.007 | 169.4±2.4 | 0.721±0.001 |
| | $g$ | **VAE** Kingma & Welling (2014) | 0.000 ±0.000 | 0.0 ±0.0 | 94.3±2.8 | 0.770 ±0.002 | 0.276 ±0.001 | 0.799±0.002 | 177.2±3.2 | 0.743±0.002 |
| | $g$ | $\beta$-**VAE** Burgess et al. (2017) | 0.088 ±0.051 | 1.6 ±0.8 | 99.8±2.4 | 0.699 ±0.001 | 0.281 ±0.001 | 0.801±0.001 | 183.3±3.3 | 0.769±0.003 |
| | $g$ | **InfoVAE** Zhao et al. (2019) | 0.000 ±0.000 | 0.0 ±0.0 | 77.8±1.6 | 0.757 ±0.003 | 0.134 ±0.001 | 0.829±0.003 | 160.7±2.5 | 0.814±0.006 |
| | $g$ | **DiffAE** Preechakul et al. (2022) | 0.155 ±0.010 | 2.0 ±0.0 | 22.7±2.1 | 0.799 ±0.002 | 0.196 ±0.001 | 0.899±0.001 | 32.1±1.1 | 0.859±0.002 |
| | $g$ | **InfoDiffusion** Wang et al. (2023) | 0.299 ±0.006 | 3.0 ±0.0 | 23.8±1.6 | 0.848 ±0.001 | 0.342 ±0.002 | 0.882±0.001 | 32.4±1.8 | 0.886±0.004 |
| $\mathcal{P}$ | $c$ | **ParamReL** ($\gamma = 1, \lambda = 0.01$) | 0.261 ±0.01 | **5.0 ±0.0** | 22.6±1.2 | 0.846 ±0.009 | 0.477 ±0.002 | 0.901±0.007 | 31.8±1.1 | 0.892±0.004 |
| | $d$ | **ParamReL** ($\gamma = 0.9, \lambda = 0.01$) | 0.302 ±0.005 | 4.0 ±0.0 | 22.1±1.6 | 0.850 ±0.006 | **0.567 ±0.005** | 0.902±0.001 | 31.2±1.1 | 0.901±0.001 |
| | $d$ | **ParamReL** ($\gamma = 1, \lambda = 0.01$) | **0.368 ±0.005** | 3.0 ±0.0 | **21.6±1.1** | **0.865±0.004** | 0.485 ±0.009 | **0.931±0.001** | 31.1±1.1 | **0.911±0.002** |

**Smooth Representation Learning for Latent Interpolation** Latent space interpolation (Goodfellow et al., 2014; Higgins et al., 2017) is commonly used to validate the smoothness, continuity, and semantic coherence of the learned latent semantics in generative models. Typically, two samples are embedded into the latent space, and interpolating between the latent variables generates interpolated representations. The reconstructed outputs produced by the sampling process reveal the semantic richness of the latent space. Demonstration of the image interpolation is detailed in Appendix D.1.

As shown in Figure 14 in Appendix E.3, ParamReL achieves near-exact reconstruction, in contrast to the downgraded performance of VAE variants such as (a) vanilla VAE, and (b) $\beta$-VAE. Compared with diffusion models (c) DiffAE and (d) InfoDiffusion, ParamReL characterizes a smoother and more consistent latent space with high-quality samples.

**Disentanglement** We perform latent traversals on the FFHQ and CelebA datasets to evaluate the disentanglement properties of our trained ParamReL, as illustrated in Figure 5 and Figure 15 in Appendix E.3. In this process, we modify one dimension of the learned latent semantics $\{\mathbf{z}_t\}_{t=1}^{T}$ each step, and replace it with $M$ evenly distributed numbers within a standardized range (e.g., $-3$ to $+3$), while keeping the other dimensions fixed. After decoding these adjusted latent semantics,

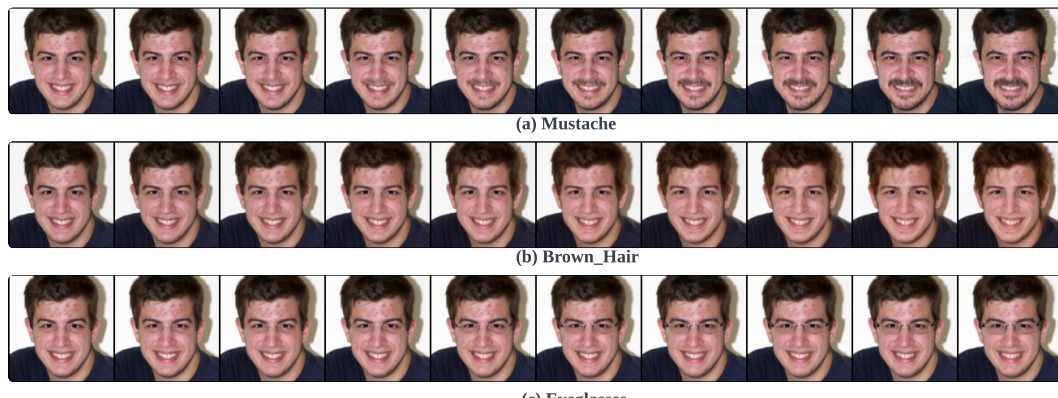

(a) Mustache

(b) Brown_Hair

(c) Eyeglasses

Figure 5: Disentanglement of ParamReL on FFHQ-128. The interpretable traversal directions are displayed by traversing the encodings ranging from $[-3, 3]$.

we evaluate the generated samples for changes in specific attributes. Successful disentanglement is verified when manipulating one single dimension alters only one distinguishable attribute, such as age, while leaving all other attributes unchanged. As shown in Figure 5 and Figure 15 in the Appendix, ParamReL effectively isolates and controls individual data attributes in both FFHQ and CelebA. For example, on FFHQ, manipulating latent dimensions controls attributes like `Mustache`, `Brown Hair`, and `Eyeglasses`, while other attributes remain constant. Similarly, on CelebA, attributes such as `Smiling`, `Pale Skin`, and `Big Nose` are independently manipulated without affecting others.

To provide a thorough and unbiased quantitative assessment of disentanglement, we utilize two metrics: 1) Disentanglement, Completeness, and Informativeness (DCI) (Eastwood & Williams, 2018), which is a prediction-based indicator; and 2) Total AUROC Difference (TAD) (Yeats et al., 2022), an intervention-based criterion. Additionally, we report the generation quality in Appendix E.3 and conclude that ParamReL achieves near-exact reconstruction on CelebA (Figure 16 (a)), Shapes3D (Figure 16 (b)), and CIFAR-10 (Figure 16 (c)). Both the qualitative latent traversal results and the quantitative disentanglement metrics show that ParamReL effectively learns disentangled representations, with visual traversals closely aligning with the attributes that the latent semantics are intended to capture.

## 6 CONCLUSION AND LIMITATIONS

In this work, we introduce ParamReL, a novel unified parameter space representation learning framework, as a unified strategy to handle continuous, discrete and even discretized data. Unlike traditional encoder methods that map observations into static latent semantics, ParamReL employs a self-encoder to derive progressively structured latent semantics from intermediate parameters at each step of the generation process. This allows for more effective representation learning across different data types. Our experiments on tasks including latent interpolation, disentanglement, time-varying conditional reconstruction, and conditional generation validate the effectiveness of ParamReL. The results demonstrate its superior ability to extract meaningful high-level semantics, leading to unified representations and a clear semantic understanding of the underlying data.

While ParamReL shows promising results, our experiments reveal areas for potential expansion. (1) The precision variables, which play a key role in the sampling process, could be further optimized to reduce computational time and to improve efficiency. This was observed during the sampling stages where slight inefficiencies in parameter updates are detected. (2) We noticed that employing a standard U-Net architecture without pre-training may limit the performance of ParamReL, particularly in tasks involving complex data. Therefore, exploring the integration of a pre-trained U-Net model into ParamReL could provide a significant boost in accuracy and representation quality. We will investigate these in the future work.

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

CONTENTS

# A   PRELIMINARIES

## A.1   BAYESIAN FLOW NETWORKS

In Graves et al. (2023), BFNs assume two types of distributions: a simple *input distribution* $P_{\mathrm{I}}(\cdot)$ representing the initial belief about observations and an *output distribution* $P_{\mathrm{O}}(\cdot)$ simulating the observation distribution. The parameters of input distribution are first updated through a Bayesian inference scheme and then passed into a neural network $\psi(\cdot)$ to form the parameters of output distributions. The main objective of BFNs is to minimize the divergence between the ground-truth data distribution and the output distribution, ensuring that the output distribution closely approximates the ground-truth data distribution.

Following the notations in diffusion models, we denote $\mathbf{x}_0$ as the observations. There are $T$ *reverse* steps in BFNs which gradually reveals the information of $\mathbf{x}_0$ through $\{\mathbf{x}_T, \mathbf{x}_{T-1}, \ldots, \mathbf{x}_1\}$ to the input distribution[2]. At each step $t$, $\mathbf{x}_t$ is first noised through a *sender distribution* $p_{\mathrm{S}}(\widehat{x}_t \,|\, \mathbf{x}_t; \alpha_t)$, with $\alpha_t$ denoting the precision. Combined with input distribution $p_{\mathrm{I}}(\mathbf{x}_t; \boldsymbol{\theta}_{t+1})$, the posterior distribution of $\mathbf{x}_t$ is obtained as $p(\mathbf{x}_t; h(\boldsymbol{\theta}_{t+1}, \widehat{x}_t, \alpha_t)) \propto p_{\mathrm{I}}(\mathbf{x}_t; \boldsymbol{\theta}_{t+1}) p_{\mathrm{S}}(\widehat{x}_t \,|\, \mathbf{x}_t; \alpha_t)$, where $\boldsymbol{\theta}_t = h(\boldsymbol{\theta}_{t+1}, \widehat{x}_t, \alpha_t)$ is the Bayesian update function. By feeding this intermediate (posterior) parameter $\boldsymbol{\theta}_t$ into a neural network $\psi(\cdot)$, $\mathbf{x}_t$'s output distribution $p_{\mathrm{O}}(\cdot)$ is parameterized as $p_{\mathrm{O}}(\mathbf{x}_t; \psi(\boldsymbol{\theta}_t))$. Finally, a *receiver distribution* $p_{\mathrm{R}}(\cdot)$ is defined as the expectation of the sender distribution with respect to the output distribution, i.e., $p_{\mathrm{R}}(\widehat{x}_t; \psi(\boldsymbol{\theta}_t), \alpha_t) := \mathbb{E}_{p_{\mathrm{O}}(\mathbf{x}_t; \psi(\boldsymbol{\theta}_t))}[p_{\mathrm{S}}(\widehat{x}_t \,|\, \mathbf{x}_t; \alpha_t)]$. See Figure 6 (a) for a visualization of the relationships between these distributions.

In BFNs, the joint distribution over the observation $\mathbf{x}_0$ and the intermediates $\{\mathbf{x}_t\}_t$ is defined as $p(\mathbf{x}_0, \{\mathbf{x}_t\}_t | -) := p_{\mathrm{O}}(\mathbf{x}_0; \psi(\boldsymbol{\theta}_0)) \prod_{t=1}^{T} p_{\mathrm{R}}(\widehat{x}_t; \psi(\boldsymbol{\theta}_t), \alpha_t)$. This intractable joint distribution can be approximated under the variational inference framework as follows:

$$\log p(\mathbf{x}_0) \geq \mathbb{E}_{p_{\mathrm{F}}(\boldsymbol{\theta}_{1:T}|-)p_{\mathrm{S}}(\{\mathbf{x}_t\}_t|-)} \left[ \log \frac{p_{\mathrm{O}}(\mathbf{x}_0; \psi(\boldsymbol{\theta}_0)) \prod_{t=1}^{T} p_{\mathrm{R}}(\widehat{x}_t; \psi(\boldsymbol{\theta}_t), \alpha_t)}{\prod_{t=1}^{T} p_{\mathrm{S}}(\widehat{x}_t \,|\, \mathbf{x}_t; \alpha_t)} \right]$$

$$= -\sum_{t=1}^{T} \underbrace{\mathbb{E}_{p_F(\boldsymbol{\theta}_t|-)} \mathrm{KL} \left[ p_{\mathrm{S}}\left(\widehat{x}_t \,|\, \mathbf{x}_0; \alpha_{T:t}\right) \,\|\, p_{\mathrm{R}}\left(\widehat{x}_t; \psi(\boldsymbol{\theta}_t), \alpha_t\right) \right]}_{\mathcal{L}_t^{\mathrm{R}}(\mathbf{x})} + \underbrace{\mathbb{E}_{p_F(\boldsymbol{\theta}_0|-)} \ln p_{\mathrm{O}}(\mathbf{x}_0; \psi(\boldsymbol{\theta}_0))}_{\mathcal{L}^{\mathrm{D}}(\mathbf{x})},$$

(4)

where $p_{\mathrm{F}}(\boldsymbol{\theta}_t|-)$ is the distribution of $\boldsymbol{\theta}_t$ (see Appendix A.2 for a detailed calculation). Maximizing Eq. 4 equals minimizing the discrepancy $\mathcal{L}_t^{\mathbf{R}}(\mathbf{x})$ between the sender and receiver distributions and penalizing Distortion $\mathcal{L}^{\mathbf{D}}(\mathbf{x})$ to maximize the likelihood distribution over data.

Table 2: Examples of detailed distribution formats in BFNs. $\boldsymbol{\theta}_{t+1} = \{\mu_{t+1}, \rho_{t+1}^{-1}\}$). cate: categorical distribution.

| Data type | $p_{\mathrm{I}}(\mathbf{x}_t|\boldsymbol{\theta}_{t+1})$ | $p_{\mathrm{S}}(\widehat{x}_t|\mathbf{x}_t;\alpha_t)$ | $\boldsymbol{\theta}_t = h(\boldsymbol{\theta}_{t+1}, \widehat{x}_t, \alpha_t)$ |
|---|---|---|---|
| Continuous data | $\mathcal{N}(\mathbf{x}_t; \mu_{t+1}, \rho_{t+1}^{-1})$ | $\mathcal{N}(\widehat{x}_t; \mathbf{x}, \alpha_t^{-1})$ | $\mu_t = \frac{\alpha_t \widehat{x}_t + \rho_{t+1}\mu_{t+1}}{\alpha_t + \rho_{t+1}}$ |
| Discrete data | $\mathrm{Cat}(\mathbf{x}_t; \frac{1}{K} \cdot \mathbf{1})$ | $\mathcal{N}(\widehat{x}_t; \alpha_t K \mathbf{e}_{\mathbf{x}_t} - \alpha_t, \alpha_t K \mathbf{I})$ | $\boldsymbol{\theta}_t = \frac{e^{\widehat{x}_t}\boldsymbol{\theta}_{t+1}}{\sum_k e^{\mathbf{x}_{t-1,k}}\boldsymbol{\theta}_{t+1,k}}$ |
| **Data type** | $p_{\mathrm{O}}(\mathbf{x}_t|\boldsymbol{\theta}_t)$ | $p_{\mathrm{R}}(\widehat{x}_t|\psi(\boldsymbol{\theta}_t), \alpha_t)$ | |
| Continuous data | $\delta(\mathbf{x}_t - \psi(\boldsymbol{\theta}_t))$ | $\mathcal{N}(\widehat{x}_t; \psi(\boldsymbol{\theta}_t), \alpha_t^{-1})$ | |
| Discrete data | $\mathrm{Cat}(\mathrm{softmax}(\psi(\boldsymbol{\theta}_t)))$ | $\sum_k p_O(k; \psi(\boldsymbol{\theta}_t))\mathcal{N}(\widehat{x}_t; \alpha_t K\mathbf{e}_k - \alpha_t, \alpha_t K\mathbf{I})$ | |

---

[2]It is noted that the index $t$ is used reversely in Graves et al. (2023). We make such changes to be consistent with the diffusion model settings Ho et al. (2020); Song et al. (2021).

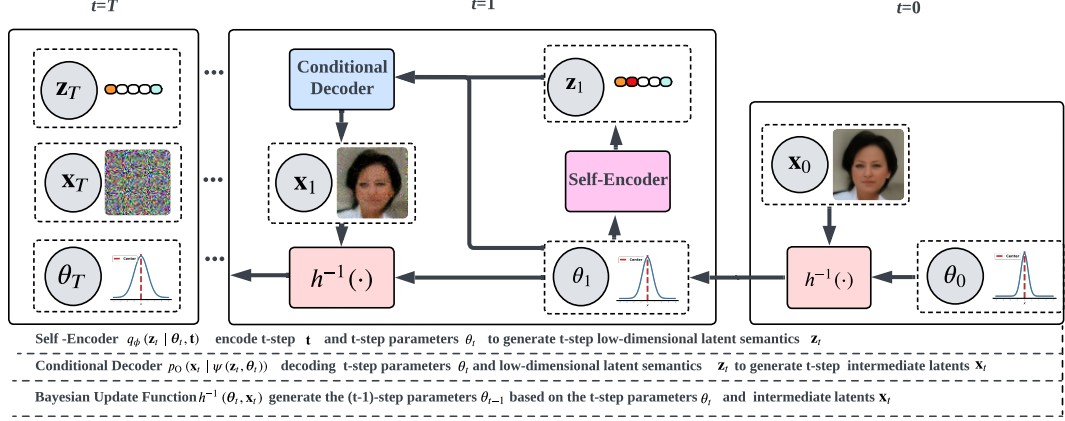

| $p_I(\mathbf{x}_t \mid \theta_{t+1})$ | $p_S(\mathbf{x}_{t-1} \mid \mathbf{x}_t; \alpha_t)$ | $p_R(\mathbf{x}_{t-1} \mid \psi(\theta_t), \alpha_t)$ | | $p_I(\mathbf{x}_t \mid \theta_{t+1})$ | $p_S(\mathbf{x}_{t-1} \mid \mathbf{x}_t; \alpha_t)$ | $p_R(\mathbf{x}_{t-1} \mid \psi(\theta_t, \mathbf{z}_t), \alpha_t)$ |

(a) Bayesian network of BFNs  (b) Bayesian network of ParamReL

Figure 6: The relationships between distributions in BFNs (a) and ParamReL (b).

Self-Encoder $q_\phi(\mathbf{z}_t \mid \theta_t, \mathbf{t})$ encode t-step **t** and t-step parameters $\theta_t$ to generate t-step low-dimensional latent semantics $\mathbf{z}_t$

Conditional Decoder $p_O(\mathbf{x}_t \mid \psi(\mathbf{z}_t, \theta_t))$ decoding t-step parameters $\theta_t$ and low-dimensional latent semantics $\mathbf{z}_t$ to generate t-step intermediate latents $\mathbf{x}_t$

Bayesian Update Function $h^{-1}(\theta_t, \mathbf{x}_t)$ generate the (t-1)-step parameters $\theta_{t-1}$ based on the t-step parameters $\theta_t$ and intermediate latents $\mathbf{x}_t$

Figure 7: The reverse-sampling process in ParamReL.

## A.2 BAYESIAN FLOW DISTRIBUTION

Bayesian flow distribution $p_F(\cdot \mid \mathbf{x}; t)$ is the marginal distribution over input parameters at time $t$, given prior distribution, accuracy schedule $\alpha$ and Bayesian update distribution $p_U(\cdot \mid \boldsymbol{\theta}, \mathbf{x}; \alpha)$, as follows:

$$p_F(\boldsymbol{\theta} \mid \mathbf{x}; t) = p_U(\boldsymbol{\theta} \mid \boldsymbol{\theta}_0, \mathbf{x}; \beta(t)). \tag{5}$$

## A.3 GENERATIVE LATENT VARIABLE MODELS FOR REPRESENTATION LEARNING

Latent Variable Models (LVMs) Everett (2013) which aim at learning the joint distribution $p(\mathbf{x}, \mathbf{z})$ over data $\mathbf{x}$ and latent variables $\mathbf{z}$ present efficient ways for uncovering hidden semantics. In LVMs, the joint distribution $p(\mathbf{x}, \mathbf{z})$ is usually decomposed as $p(\mathbf{x}, \mathbf{z}) = p(\mathbf{x} \mid \mathbf{z})p(\mathbf{z})$, where $p(\mathbf{z})$ represents prior knowledge for inference Tschannen et al. (2018), thus facilitating learning the conditional distribution $p(\mathbf{x} \mid \mathbf{z})$. Among LVMs, Variational AutoEncoders (VAEs) Kingma & Welling (2014) and diffusion models Ho et al. (2020); Song et al. (2021) are two representative approaches Kwok & Adams (2012).

In VAEs, latent variables $\mathbf{z}$ is obtained through an *encoder network* $q_\phi(\mathbf{z} \mid \mathbf{x})$, whereas observations are reconstructed through a *decoder network* $p_\theta(\mathbf{x} \mid \mathbf{z})$, with $\phi$ and $\theta$ being the encoder and decoder parameters.

The dimensions of $\mathbf{z}$ are usually much smaller than those of $\mathbf{x}$, denoted as $|\mathbf{z}| \ll |\mathbf{x}|$, such that redundant information is effectively removed and the most semantically meaningful factors are abstracted Louizos et al. (2016). VAEs are popular for downstream tasks like disentanglement Higgins et al. (2017); Yang et al. (2022); Hwang et al. (2023); Esmaeili et al. (2023), classification Takahashi et al. (2022); Tonekaboni et al. (2022), and clustering Jiang et al. (2016); Xu et al. (2021).

On the other hand, diffusion models Ho et al. (2020); Song et al. (2021) first use $T$ diffusion steps to transform observation $\mathbf{x}$ into a white noise $\mathbf{x}_T$ and then use $T$ denoising steps to reconstruct the observation. Diffusion models have obtained impressive performance in the fidelity and diversity of generation tasks. However, they might be unable to obtain meaningful latent semantics since the dimensions of $\mathbf{x}$ and $\mathbf{x}_T$ are the same as $|\mathbf{x}| = |\mathbf{x}_T|$. Preechakul et al. (2022); Wang et al. (2023)

have attempted to integrate a decodable auxiliary variable $\mathbf{z}$ to enable diffusion models to obtain low-dimensional latent semantics. However, they have not overcome issues like the slow training speed inherent to the diffusion and reverse processes.

## A.4 ILLUSTRATION OF PARAMETER SPACE OPTIMIZATION

Figure 8 illustrates the optimal data distribution learned in the parameter space. The plot presents stochastic parameter trajectories for the input distribution mean (indicated by white lines) overlaid on a Bayesian flow distribution logarithmic heatmap.

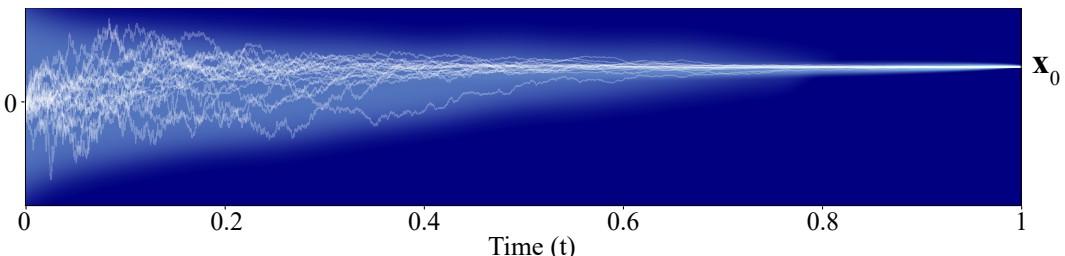

Figure 8: This figure illustrates optimization in the parameter space after $t$ iterations.

## B PROOFS

### B.1 DERIVATION OF ELBO FOR PARAMREL

We derive the ELBO of ParamReL defined in Eq. (3).

$$
\log p(\mathbf{x}_0)
$$

$$
= \log \int_{\{\mathbf{z}_t\}_t} \int_{\{\mathbf{x}_t\}_t} p\left(\mathbf{x}_0, \{\mathbf{x}_t\}_t, \{\mathbf{z}_t\}_t \mid \boldsymbol{\theta}_0, \alpha\right) \mathrm{d}\{\mathbf{z}_t\}_t \mathrm{d}\{\mathbf{x}_t\}_t
$$

$$
= \log \int_{\{\boldsymbol{\theta}_t\}_t} \int_{\{\mathbf{z}_t\}_t} \int_{\{\mathbf{x}_t\}_t} p(\{\boldsymbol{\theta}_t\}_t|-) p_O(\mathbf{x}_0; \psi(\boldsymbol{\theta}_0, \mathbf{z}_0)) \prod_{t=T}^{1} p(\mathbf{z}_t) \mathbb{E}_{p_O(\mathbf{x}_t; \psi(\boldsymbol{\theta}_t, \mathbf{z}_t))}[p_S(\mathbf{x}_{t-1} \mid \mathbf{x}_t; \alpha_t)]
$$

$$
\mathrm{d}\{\mathbf{z}_t\}_t \mathrm{d}\{\mathbf{x}_t\}_t \mathrm{d}\{\boldsymbol{\theta}_t\}_t
$$

$$
= \log \int_{\{\mathbf{z}_t\}_t} \int_{\{\mathbf{x}_t\}_t} \int_{\{\boldsymbol{\theta}_t\}_t} p(\{\boldsymbol{\theta}_t\}_t|-) \frac{p_O(\mathbf{x}_0; \psi(\boldsymbol{\theta}_0, \mathbf{z}_0)) \prod_{t=T}^{1} p(\mathbf{z}_t) \mathbb{E}_{p_O(\mathbf{x}_t; \psi(\boldsymbol{\theta}_t, \mathbf{z}_t))}[p_S(\mathbf{x}_{t-1} \mid \mathbf{x}_t; \alpha_t)]}{\prod_{t=1}^{T} p_S(\mathbf{x}_{t-1} \mid \mathbf{x}_t; \alpha_t) q_\phi(\mathbf{z}_t | \boldsymbol{\theta}_t, t)}
$$

$$
\cdot \prod_{t=1}^{T} p_S(\mathbf{x}_{t-1} \mid \mathbf{x}_t; \alpha_t) q_\phi(\mathbf{z}_t | \boldsymbol{\theta}_t, t) \mathrm{d}\{\mathbf{z}_t\}_t \mathrm{d}\{\mathbf{x}_t\}_t \mathrm{d}\{\boldsymbol{\theta}_t\}_t
$$

$$
\geq \mathbb{E}_{\prod_{t=1}^{T} p_S(\mathbf{x}_{t-1} \mid \mathbf{x}_t; \alpha_t) q_\phi(\mathbf{z}_t | \boldsymbol{\theta}_t, t) p(\boldsymbol{\theta}_t|-)} \left[ \log \frac{p_O(\mathbf{x}_0; \psi(\boldsymbol{\theta}_0, \mathbf{z}_0)) \prod_{t=T}^{1} p(\mathbf{z}_t) \mathbb{E}_{p_O(\mathbf{x}_t; \psi(\boldsymbol{\theta}_t, \mathbf{z}_t))}[p_S(\mathbf{x}_{t-1} \mid \mathbf{x}_t; \alpha_t)]}{\prod_{t=1}^{T} p_S(\mathbf{x}_{t-1} \mid \mathbf{x}_t; \alpha_t) q_\phi(\mathbf{z}_t | \boldsymbol{\theta}_t, t)} \right]
$$

$$
= \sum_{t=1}^{T} \mathbb{E}_{p_F(\boldsymbol{\theta}_t|-)} \mathbb{E}_{q_\phi(\mathbf{z}_t)} \left\{ \mathbb{E}_{p_S(\mathbf{x}_{t-1} \mid \mathbf{x}_0; \alpha_{T:t})} \left[ \log \frac{p_S(\mathbf{x}_{t-1} \mid \mathbf{x}_0; \alpha_{T:t})}{p_R(\mathbf{x}_{t-1}; \psi(\boldsymbol{\theta}_t, \mathbf{z}_t), \alpha_t)} \right] \right.
$$

$$
\left. - \mathbb{E}_{q_\phi(\mathbf{z}_t | \boldsymbol{\theta}_t)} \left[ \log \frac{q_\phi(\mathbf{z}_t | \boldsymbol{\theta}_t)}{p(\mathbf{z}_t)} \right] \right\} + \mathbb{E}_{q_\phi(\mathbf{z}_0, \boldsymbol{\theta}_0)} \left[ \ln p_O(\mathbf{x}_0; \psi(\boldsymbol{\theta}_0, \mathbf{z}_0)) \right]
$$

$$
= - \sum_{t=1}^{T} \mathbb{E}_{p_F(\boldsymbol{\theta}_t|-)} \mathbb{E}_{q_\phi(\mathbf{z}_t)} \left\{ \mathrm{KL} \left[ p_S(\mathbf{x}_{t-1} \mid \mathbf{x}_0; \alpha_{T:t}) \parallel p_R(\mathbf{x}_{t-1}; \psi(\boldsymbol{\theta}_t, \mathbf{z}_t), \alpha_t) \right] \right.
$$

$$
\left. - \mathrm{KL} \left[ q_\phi(\mathbf{z}_t | \boldsymbol{\theta}_t) \parallel p(\mathbf{z}_t) \right] \right\} + \mathbb{E}_{q_\phi(\mathbf{z}_0, \boldsymbol{\theta}_0)} \left[ \ln p_O(\mathbf{x}_0; \psi(\boldsymbol{\theta}_0, \mathbf{z}_0)) \right] := \mathcal{L}_{\texttt{ParamReL}} \quad (6)
$$

## B.2 Mutual Information Learning

$$\mathcal{L}_{\texttt{ParamReL+}} = -\sum_{t=1}^{T} \mathbb{E}_{p_{\text{F}}(\boldsymbol{\theta}_t|-)}\mathbb{E}_{q_\phi(\mathbf{z}_t)} \left\{ \text{KL}\left[p_{\text{S}}\left(\mathbf{x}_{t-1}\mid\mathbf{x}_0;\alpha_{T:t}\right) \ \| \ p_{\text{R}}\left(\mathbf{x}_{t-1};\psi(\boldsymbol{\theta}_t,\mathbf{z}_t),\alpha_t\right)\right] \right.$$

$$\left. -\frac{1-\gamma}{T}\text{KL}\left[q_\phi\left(\mathbf{z}_t\mid\boldsymbol{\theta}_t\right) \ \| \ p(\mathbf{z})\right] - \frac{\gamma+\lambda-1}{T}\text{KL}\left[q_\phi\left(\mathbf{z}_t\right) \ \| \ p(\mathbf{z})\right] \right\} + \mathbb{E}_{q_\phi(\mathbf{z}_0,\boldsymbol{\theta}_0)}\left[\ln p_{\text{O}}(\mathbf{x}_0;\psi(\boldsymbol{\theta}_0,\mathbf{z}_0))\right].$$

$$(7)$$

Unlike the rest of the terms that can be optimized directly using reparameterization tricks, the TC term cannot be directly optimized due to intractable marginal distribution $q_\phi(\mathbf{z}_t)$. Here, we follow the guidance in Zhao et al. (2019) to replace the TC term with any strict divergence $D$, where $D\left(q_\phi(\mathbf{z})\|p(\mathbf{z})\right) = 0$ iff $q_\phi(\mathbf{z}) = p(\mathbf{z})$. We implement the Maximum-Mean Discrepancy (MMD) Zhao et al. (2019) from the divergence family. MMD is a statistical measure that quantifies the difference between two probability distributions by comparing their mean embeddings in a high-dimensional feature space. By defining the kernel function $\kappa(\cdot,\cdot)$, $D_{\text{MMD}}$ is denoted as:

$$D_{\text{MMD}}\left(q(\cdot)\|p(\cdot)\right) = \mathbb{E}_{p(\mathbf{z}),p(\mathbf{z}')}\left[\kappa\left(\mathbf{z},\mathbf{z}'\right)\right] - 2\mathbb{E}_{q(\mathbf{z}),p(\mathbf{z}')}\left[\kappa\left(\mathbf{z},\mathbf{z}'\right)\right] + \mathbb{E}_{q(\mathbf{z}),q(\mathbf{z}')}\left[\kappa\left(\mathbf{z},\mathbf{z}'\right)\right]. \quad (8)$$

# C Technical Details and Experimental Setup

## C.1 Encoder Architecture

In our proposed encoder architecture, the self-encoder $q_\phi(\mathbf{z}_t|\boldsymbol{\theta}_t,t)$ also conditions on step $(t+1)$'s upsampling layers $\{\mathbf{u}_{t+1,l}\}_{l=1}^L$, where $L$ is the number of layers in the U-Net architecture. For the $l$-th upsampling layer $\mathbf{u}_{t+1,l}$ at step $t+1$, we upsample it to the size of $\mathbf{x}_t$, update by the Bayesian update function, and pass through a bottleneck layer $B_l(\cdot)$ (He et al., 2016) to the low-dimensional size. As a result, the self-encoder is defined as $q_\phi(\mathbf{z}_t|\boldsymbol{\theta}_t,t) = \mathcal{N}\left(\mathbf{z}_t; g_\mu(\boldsymbol{\theta}_t, \{\mathbf{u}_{t+1,l}\}_{l=1}^L,t), g_\sigma(\boldsymbol{\theta}_t, \{\mathbf{u}_{t+1,l}\}_{l=1}^L,t)^2\right)$, where $g_\mu(\cdot), g_\sigma(\cdot)$ use the same structure as:

$$g_\mu(\boldsymbol{\theta}_t, \{\mathbf{u}_{t+1,l}\}_{l=1}^L,t), g_\sigma(\boldsymbol{\theta}_t, \{\mathbf{u}_{t+1,l}\}_{l=1}^L,t) := \sum_{l=0}^{L} \omega_l \cdot B_l(h(\mathbf{x}_t, \mathbf{u}_{t+1,l})) + \omega_{L+1} \cdot B_{L+1}(\boldsymbol{\theta}_t)$$

where $\omega_l$ is the mixing weight of the $l$-th layer.

## C.2 BFN Architecture

Similar to the diffusion-based representation learning model, we update the U-Net architecture based on Residual Blocks and Attention Modules. However, unlike previous approaches Ho et al. (2020); Song et al. (2021); Preechakul et al. (2022); Wang et al. (2023), we use shallower layers in the upper and down modules while incorporating an additional attention mechanism in the bottleneck module to achieve significant representations. Figure 9 illustrates the specific structural differences.

## C.3 Hyperparameters

Table 3 presents the hyperparameter settings for training ParamReL. Different bin values are provided for various continuous datasets. All models are trained for 50 epochs. "Channel mult" denotes the channel shapes in each ResNet block within the U-Net architecture.

# D Experiment details

## D.1 Interpolation

The latent space interpolation can be described as follows. Firstly, we noise source images to generate latent pairs by sender distribution, $< \mathbf{x}_1^1, \mathbf{x}_1^2 >$, where $\mathbf{x}_1^1 \sim q(\cdot \mid \mathbf{x}_N^1)$ and $\mathbf{x}_1^2 \sim q(\cdot \mid \mathbf{x}_N^2)$.

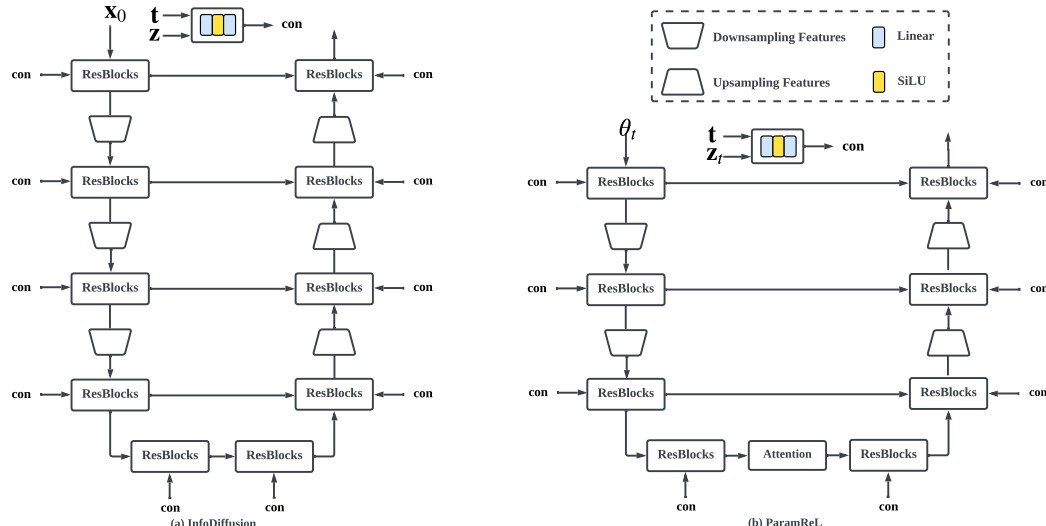

Figure 9: U-Net comparisons of InfoDiffusion (a) and ParamReL (b). We apply the Attention module in the bottleneck layer, shallower than the InfoDiffusion's U-Net.

Table 3: Hyperparameters for training Bayesian Flow Networks, U-Net architecture, training protocols, and devices. The training configuration of ParamReL is based on Preechakul et al. (2022); Dhariwal & Nichol (2021).

|  | Hyperparameter | CelebA | Shapes3D | CIFAR-10 | FFHQ |
|---|---|---|---|---|---|
| **Encoder** | Encoder base channels | 64 | 64 | 64 | 128 |
|  | Encoder attention resolution | [16] | [16] | [16] | [16] |
|  | Encoder channel multipliers | [1,2,4,8,8] | [1,1,2,3,4,4] | [1,1,2,3,4,4] | [1,1,2,3,4,4] |
|  | Latent code $\mathbf{z}$ dimension | 512 | 512 | 512 | 512 |
| **Decoder** | Base channels | 64 | 64 | 64 | 128 |
|  | Channel multipliers | [1,2,4,8] | [1,1,2,3,4] | [1,1,2,3,4] | [1,1,2,3,4] |
|  | Attention resolution | [16] | [16] | [16] | [16] |
|  | Images trained | 130M | 130M | 130M | 130M |
|  | Batch size | 128 | 128 | 128 | 128 |
|  | Learning rate | | 1e-4) | | |
|  | Optimizer | | Adam (no weight decay) | | |
|  | EMA rate | | 0.9999 | | |
|  | Training $T$ | | 1000 | | |
|  | Diffusion loss | | MSE with noise prediction $\epsilon$ | | |
|  | Diffusion var. | | Not important for DDIM | | |
| **Device** | GPU | H100 | H100 | H100 | H100 |

Then, we implement two methods from Shoemake (1985) to generate four interpolated latent pairs $\bar{\mathbf{x}}_{1:4}$, i.e., linear interpolation, and spherical interpolation:

$$\bar{\mathbf{x}}_i = (1-\lambda)\mathbf{x}_0^1 + \lambda\mathbf{x}_0^2,$$
$$\bar{\mathbf{x}}_i = \frac{\sin((1-\alpha)\theta)}{\sin(\theta)}\mathbf{x}_0^1 + \frac{\sin(\alpha\theta)}{\sin(\theta)}\mathbf{x}_0^1, \tag{9}$$

where $\lambda$ is the scale coefficient, $\alpha \in [0,1]$ denotes the interpolation steps, and $\theta = \arccos\left(\frac{\left(\mathbf{x}_0^1\right)^\top \mathbf{x}_0^2}{\|\mathbf{x}_0^1\|\|\mathbf{x}_0^2\|}\right)$ is the angle between $\mathbf{x}_0^1$ and $\mathbf{x}_0^2$.

Table 4: Comparison of representation learning algorithms on continuous data by disentanglement performance (mean ± std) and classification. The quantitative results for each algorithm averaged over five trials. (Modeling on data space $\mathcal{D}$, parameter space $\mathcal{P}$; Prior distribution specify: Gaussian $g$, Categorical $c$, Delta $d$; ↑ higher is better, ↓ lower is better; [**Top-1**, Top-2, Top-3]).

| Prior on | Prior type | Methods | CelebA | | | | 3DShapes | | CIFAR-10 | |
|---|---|---|---|---|---|---|---|---|---|---|
| | | | $\mathcal{TAD}$ ↑ | $\mathcal{ATTRS}$ ↑ | $\mathcal{FID}$ ↓ | $\mathcal{AUROC}$ ↑ | $\mathcal{DCI}$ ↑ | $\mathcal{AUROC}$ ↑ | $\mathcal{FID}$ ↓ | $\mathcal{AUROC}$ ↑ |
| | - | **AE** | 0.042 ±0.004 | 1.0 ±0.0 | 90.4±1.8 | 0.759 ±0.003 | 0.219 ±0.001 | 0.796±0.007 | 169.4±2.4 | 0.721±0.001 |
| | $g$ | **VAE** Kingma & Welling (2014) | 0.000 ±0.000 | 0.0 ±0.0 | 94.3±2.8 | 0.770 ±0.002 | 0.276 ±0.001 | 0.799±0.002 | 177.2±3.2 | 0.743±0.002 |
| $\mathcal{D}$ | $g$ | **$\beta$-VAE** Burgess et al. (2017) | 0.088 ±0.051 | 1.6 ±0.8 | 99.8±2.4 | 0.699 ±0.001 | 0.281 ±0.001 | 0.801±0.001 | 183.3±3.1 | 0.769±0.003 |
| | $g$ | **InfoVAE** Zhao et al. (2019) | 0.000 ±0.000 | 0.0 ±0.0 | 77.8±1.6 | 0.757 ±0.003 | 0.134 ±0.001 | 0.829±0.003 | 160.7±2.5 | 0.814±0.006 |
| | $g$ | **DiffAE** Preechakul et al. (2022) | 0.155 ±0.010 | 2.0 ±0.0 | 22.7±2.1 | 0.799 ±0.002 | 0.196 ±0.001 | 0.899±0.001 | 32.1±1.1 | 0.859±0.002 |
| | $g$ | **InfoDiffusion** Wang et al. (2023) | 0.299 ±0.006 | 3.0 ±0.0 | 23.8±1.6 | 0.848 ±0.001 | 0.342 ±0.002 | 0.882±0.001 | 32.4±1.8 | 0.886±0.004 |
| | $c$ | **ParamReL** ($\gamma = 0.9, \lambda = 0.1$) | 0.221±0.032 | 3.0 ±0.0 | 23.8±1.7 | 0.841 ±0.006 | 0.453 ±0.002 | 0.871±0.007 | 33.6±2.3 | 0.857±0.005 |
| | $c$ | **ParamReL** ($\gamma = 0.9, \lambda = 0.01$) | 0.286±0.001 | 4.0 ±0.0 | 24.7±1.3 | 0.848 ±0.002 | 0.477 ±0.002 | 0.892±0.006 | 33.2±0.6 | 0.871±0.002 |
| $\mathcal{P}$ | $c$ | **ParamReL** ($\gamma = 1, \lambda = 0.1$) | 0.256±0.008 | 3.0 ±0.0 | 22.5±1.2 | 0.839 ±0.003 | 0.417 ±0.002 | 0.891±0.001 | 31.9±1.1 | 0.868±0.003 |
| | $c$ | **ParamReL** ($\gamma = 1, \lambda = 0.01$) | 0.261 ±0.01 | **5.0 ±0.0** | 22.6±1.2 | 0.846 ±0.009 | 0.477 ±0.002 | 0.901±0.007 | 31.8±1.1 | 0.892±0.004 |
| | $d$ | **ParamReL** ($\gamma = 0.9, \lambda = 0.1$) | 0.299 ±0.005 | 3.0 ±0.0 | 24.1±1.1 | 0.844 ±0.012 | 0.482 ±0.001 | 0.891±0.002 | 34.7±0.9 | 0.882±0.005 |
| | $d$ | **ParamReL** ($\gamma = 0.9, \lambda = 0.01$) | 0.302 ±0.005 | 4.0 ±0.0 | 22.1±1.6 | 0.850 ±0.116 | **0.567 ±0.005** | 0.902±0.001 | 31.2±1.1 | 0.901±0.001 |
| | $d$ | **ParamReL** ($\gamma = 1, \lambda = 0.1$) | 0.287 ±0.005 | 3.0 ±0.0 | 23.6±1.7 | 0.821 ±0.006 | 0.441 ±0.008 | 0.887±0.002 | 32.8±2.1 | 0.877±0.002 |
| | $d$ | **ParamReL** ($\gamma = 1, \lambda = 0.01$) | **0.368 ±0.005** | 3.0 ±0.0 | **21.6±1.1** | **0.865±0.004** | 0.485 ±0.009 | **0.931±0.001** | 31.1±1.1 | **0.911±0.002** |

# E ADDITIOANL RESULTS

## E.1 SENSITIVITY ANALYSIS

The coefficient in the Eq. 7 will regulate the information flow under the variational bottleneck guidance Burgess et al. (2017); Shao et al. (2020); Wu et al. (2024), resulting in the tradeoff between generation and representation learning.

Figure 10 depicts the generation and representation tradeoff in discrete datasets under the different coefficient sets ($\gamma = \{0.9, 1\}, \lambda = \{0.01, 0.1\}$). When disentanglement pressure is applied (), the AUROC increases.

Table 4 depicts the generation and representation tradeoff in continous datasets under the different coefficient sets ($\gamma = \{0.9, 1\}, \gamma = \{0.1, 0.01\}$). ParamReL consistently scores highest on average, with moderate variance.

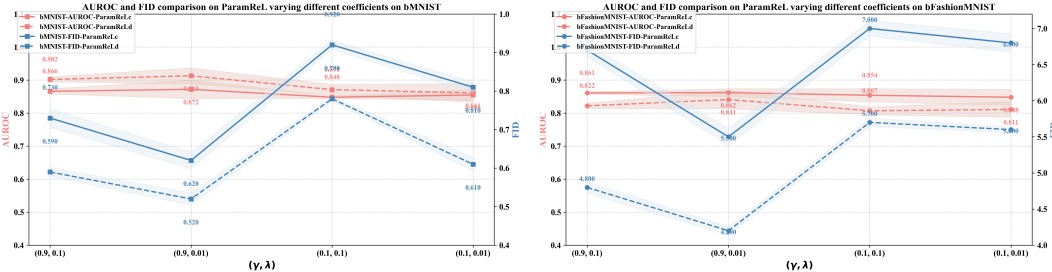

Figure 10: Effect of $\gamma$, and $\lambda$ by different representation learning metrics over ParamReLd and ParamReLc. Notations: [AUROC, FID]; [(●, bMNIST), (■, bFashionMNIST)]; [(−, ParamReLd),(− · −, ParamReLc)].

## E.2 LOW RESOLUTION REPRESENTATION LEARNING

We illustrate the representation learning ability in CelebA for high-level representation learning in Figure 13 (a), time-varying representation learning in Figure 13 (b), latent interpolation in Figure 14 and disentanglement in Figure 15.

E.3 UNCONDITIONAL GENERATION

Figure 12 illustrates the unconditional generation quality on bMNIST. Images sampled from VAE-based model are blurry, as shown in Figure 12 (b). We implement two sampling strategies in the Diffusion-based model Wang et al. (2023), and both can only sample grey-scale images. Figure 12 (c) is sampled from the DDIM sampler, and Figure 12 (d) is sampled from a two-phased sampling procedure: form timesteps $T$ to $T/2$, denoise and sample using a pre-trained vanilla denoising diffusion model. For timesteps ranging from $T/2$ to 0, proceed with sampling utilizing the InfoDiffusion model. Figure 12 (e) is images generated from our ParamReLc model. We can conclude that ParamReL can be sampled from the discrete distribution where the image value is binarized.

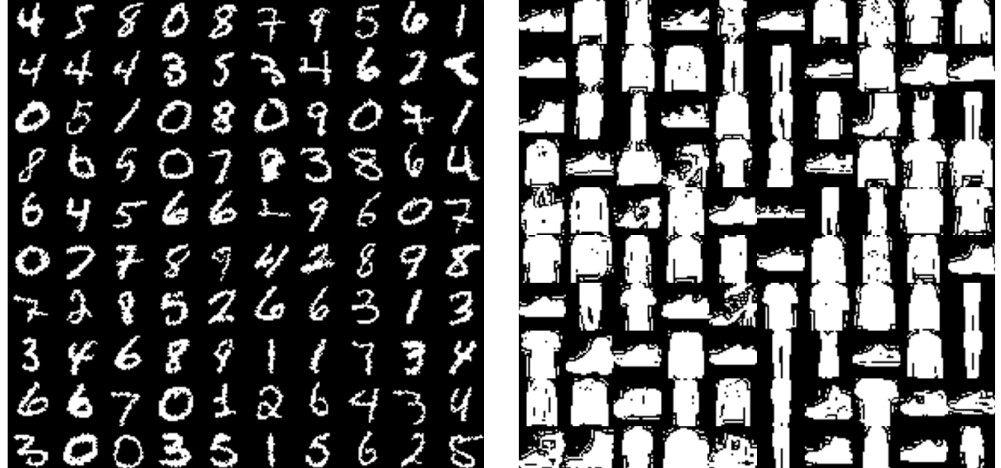

(a) : Generated samples of ParamReL on binaryMNIST (b) : Generated samples of ParamReL on binaryFashionMNIST

Figure 11: Samples reconstructed from our trained ParamReL on dataset Binary-MNIST.



(a) Binary-Mnist     (b) VAE     (c) infoDiffusion     (d) infoDiffusion(twoStage)     (e) ParamReL

Figure 12: Samples generated from our trained ParamReL.

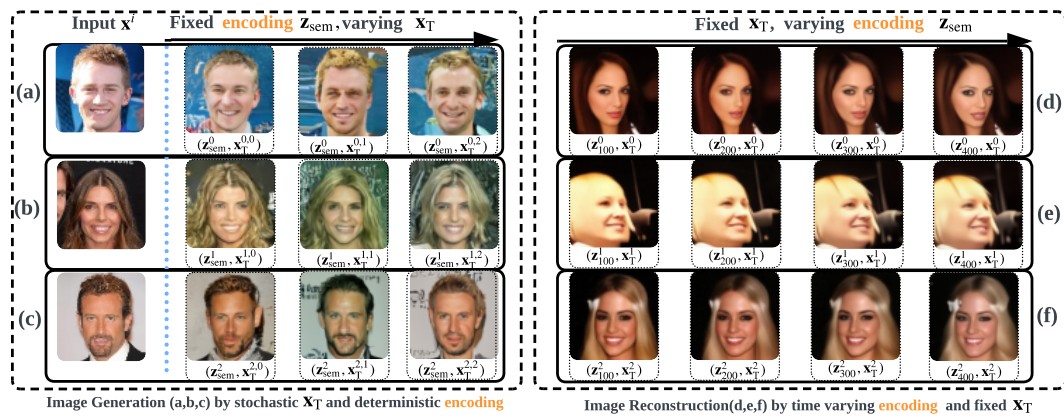

Figure 13: The left panel (a-b) shows high-level latent semantic captured by $\mathbf{z}_{\text{sem}}$ from ParamReL's encoders. By fixing $\mathbf{z}_{\text{sem}}$, the global characters of the images are invariant. By varying the stochastic $\mathbf{x}_T$, the local attributes in the corresponding generated images may vary, such as the `Narrow_Eyes` attribute in (a), the `Blond_Hair` attribute in (b), and the `Mouth_Slightly_Open` attribute in (c). The right panel (d-f) illustrates the time-varying changes that ParamReL's progressive encodes interfaced. By varying time encodes at 100, 200, 300 time steps, more attributes will be influenced in the reconstruction stage: the `Big_Lips, Pointy_Nose` attributes in (d), the `Blond_Hair, Bald` attributes in (e) and the `Wavy_Hair, High_Cheekbones` attributes in (f).

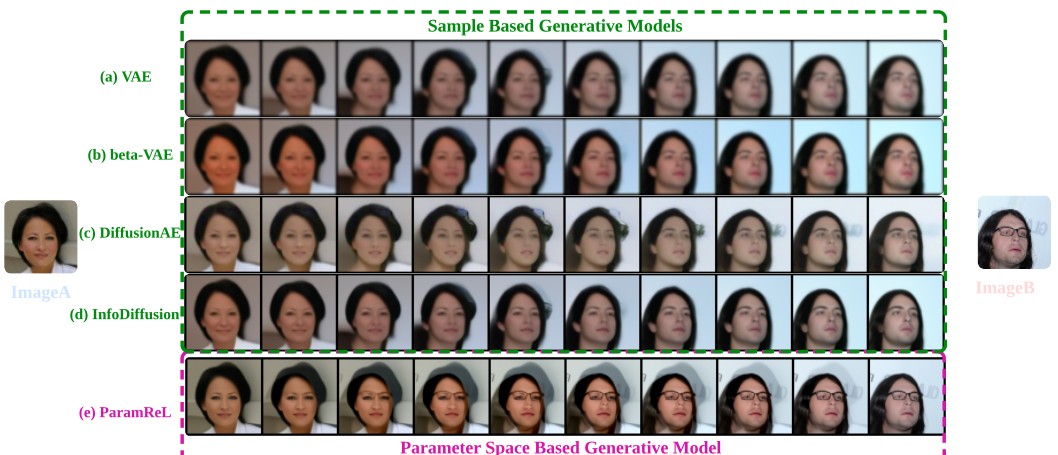

Figure 14: Comparisons of latent space interpolation among sample-based models and parameter-based models on dataset CelebA. Only our ParamReL model (e) can learn a continuous, smooth latent space while ensuring near-exact image reconstruction. Specifically, while sample-based generative models can learn a continuous but unsmooth latent space, this leads to incomplete reconstructions. For example, in (a-d), the attribute of eyeglasses is frequently omitted. Moreover, VAEs (a,b) tend to produce blurry images. Additionally, it is observable that sample-based models often compromise reconstruction in favour of representation learning, as evidenced by the failure of diffusion model variants (c-d) to accurately reconstruct background characters in imageB.

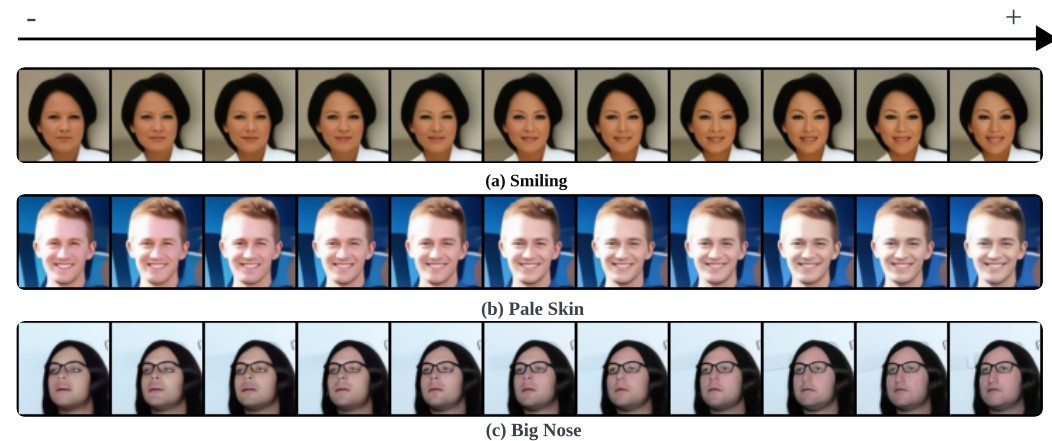

Figure 15: Disentanglement of ParamReL on CelebA. The interpretable traversal directions are displayed by traversing the encodings ranging from $[-3, 3]$.

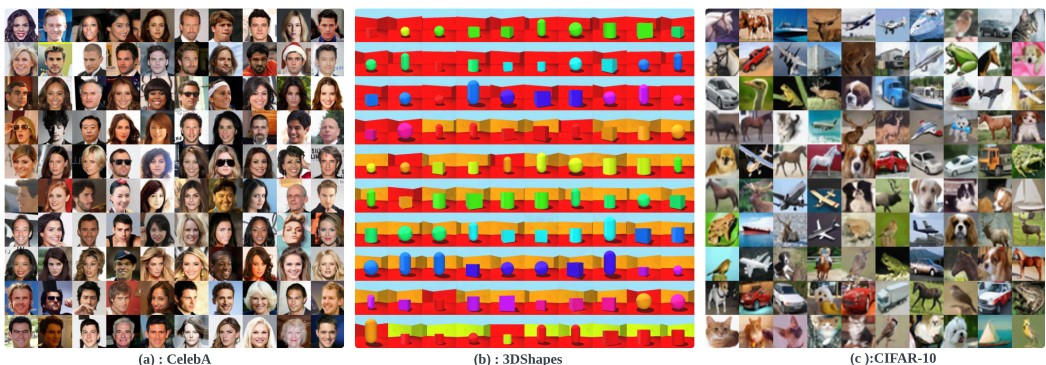

Figure 16: Generated samples trained ParamReL on CelebA (a), Shapes3D (b), CIFAR-10 (c).

