# OpenReview forum: "Parameter Space Representation Learning on Mixed-type Data"
_ICLR.cc/2025/Conference — Submitted to ICLR 2025_

### Official Review · Reviewer_ggok · 2024-10-20

**Soundness:** 3
**Presentation:** 2
**Contribution:** 2
**Rating:** 5
**Confidence:** 3

**Summary:**

The paper introduces ParamReL, a framework that aims to answer the question of how one can learn latent semantics in parameter spaces rather than observation spaces of mixed-type data. The work builds on Bayesian Flow Networks (BFNs), which can model mixed-type data by operating in the parameter space.

However, BFNs cannot capture high-level latent semantics in data. To that end, the authors additionally introduce a sequence of (low-dimensional) latent semantics $z_t$ obtained through a self-encoder and trained by maximizing a lower bound on the marginal log-likelihood.

**Strengths:**

- The proposed framework works for mixed-type data, which is often not the case for other representation learning frameworks.
- Visualizations from latent traversals indicate that ParamReL indeed learns semantics in the data
- Based on AUROC + FID, the results suggest that the method is able to learn meaningful representations while being able to reconstruct the observations
- Recap of Bayesian Flow networks and visualizations helps to i) understand the method better and ii) understand the difference between BFNs

**Weaknesses:**

- The motivation in Section 3 why mutual information regularization is needed is not clear to me. In particular, the authors mention that “Considering that we cannot optimize this object directly, we can rewrite it by factorizing the rate term into mutual information and total correlation (TC)" without explaining what the “rate term” and “TC” is. Even after reading Appendix B, it is still unclear to me.
- It would be great if the authors could add a small section in the Appendix where they define and explain the meaning of the different performance criteria used in their work.
- The results are reported for two different hyperparameters $\gamma,\lambda$. However, $\lambda$ is never introduced in the main text of the paper.  After searching the Appendix I could find it in Appendix B2 where it is introduced without mentioning why it is necessary. There is another $\lambda$ in Eq. 9), however, I doubt that it has the same meaning as in the experiment table.

**Questions:**

- In lines 236-237 the authors mention “Given the straightforward definition of Bayesian update function h(·), its inverse operation is generally easy to derive. The details of such results can be found in Figure 14.” However, Figure 14 does not provide any details on the Bayesian update function. Were the authors referring to Table 2?
- Have the authors tried to optimize the ELBO directly instead of using an additional MI term? It would be interesting to see where the former goes wrong and to justify why the MI term is actually needed.
- In Section 3.5, the caption is called “Variational Inference for Intractable Joint Distribution”. However, the joint distribution seems to be tractable, the authors even give an expression.
- How long does it take to train the method in comparison to other methods?
- What does the ATTRS column in Table 1 indicate? Is that mentioned somewhere?
- What is the influence of $\gamma,\lambda$? What happens if I set them low very low/high?

**Details Of Ethics Concerns:**

None.

---

### Official Review · Reviewer_tQLU · 2024-10-31

**Soundness:** 2
**Presentation:** 1
**Contribution:** 2
**Rating:** 3
**Confidence:** 3

**Summary:**

The authors introduce ParamReL, a parameter space representation learning framework, to enable low-dimensional latent representation learning for mixed-type data within Bayesian Flow Networks (BFNs). ParamReL incorporates a self-encoder network within BFNs to progressively encode the annealed parametric distributions of BFNs. The latents are then used to condition the BFN’s decoder. They formulate a variational learning objective regularized by a mutual information term to enhance disentangled representation learning. Experiments on binarized MNIST, binarized FashionMNIST, CelebA, CIFAR10, and Shapes3D demonstrate that ParamReL achieves high-quality sample generation and effective latent representation learning across discrete and continuous data types.

**Strengths:**

- ParamReL represents a promising new direction for defining latent semantics in mixed-type data.
- By incorporating a self-encoder within the BFN framework, the authors establish a progressive encoding process that captures latent semantics across multiple steps. This design is a contribution as it supports disentanglement and time progression within the latent space. The inclusion of a mutual information term in the variational objective is well-justified.
- The method demonstrates high-quality results across standard benchmarks, including binarized MNIST, binarized FashionMNIST, CelebA, CIFAR10, and Shapes3D.

**Weaknesses:**

- The writing would benefit from greater precision and clarity. Specific points for improvement are provided in the questions and comments. (e.g., “parameters of BFNs” vs. “parameters of BFN-produced distributions”)
- The proposed method appears closely related to infoDiffusion, with the main adaptation being its application to BFNs. It would strengthen the paper if the authors explicitly outlined the unique contributions of ParamReL, distinguishing it from infoDiffusion by clarifying any innovations that aren't due to BFNs.
- The presentation begins with the question, “How to learn latent semantics in parameter spaces rather than in observation spaces of *mixed-type data* comprising continuous, discrete, and even discretized observations?” However, the experiments are limited to discrete and continuous distributions, with no testing on mixed-type data. Additionally, the discrete data tested is exclusively binary.
- The linked GitHub repository currently includes only a README.md file, with no implementation code available.

**Questions:**

**1 Introduction**
- The authors correctly cite prior work highlighting the challenges of mixed-type data representation learning. However, explicitly mentioning real-world applications that use mixed-type data latent representations could clarify the motivation of this work. Additionally, it would help to articulate the specific advantages of parameter-space representation learning over output-space representation. (l. 37-42)

- The promise of using Bayesian Flow Networks (BFNs) as a foundation would be more compelling if the authors provided a clearer intuition on why BFNs are better suited for handling mixed-type data than diffusion models and how they refine parametric distributions. Furthermore, what limitations prevent BFNs from capturing low-dimensional latent semantics? How would a latent representation benefit the model: are we aiming to enable latent space operations, similar to those seen in VAEs and GANs, such as latent walks? (l. 43-48)

- The authors refer to “parameters of BFNs” where it might be clearer to specify “parameters of the distribution produced by BFNs” (e.g., line 53). Distinguishing these concepts would add clarity. Additionally, I thought that “parameter space representation learning” referred to learning embeddings of the model’s own parameters, which supports model transferability and interpretability. Since ParamReL instead learns representations of the parameters of the probability distribution generated by BFNs, emphasizing this distinction could help avoid reader confusion.

**2 Understanding Bayesian Flow Networks -- An Alternative View**
- The first paragraph suggests an “alternative view” of BFNs, yet it’s not immediately clear how this perspective diverges from the original one. Additionally, the point about the accessibility of BFN concepts in the original formulation might not be necessary.

- How do BFNs avoid the expressiveness limitations seen in VAEs, where the variational distribution can yield overly simplistic distributions and result in sample over-smoothing?

**3 ParamReL: Parameter Space Representation Learning**
- In lines 140-162, it is unclear which specific contributions come from Baranchuk et al. (2021), Rombach et al. (2022), and Luo et al. (2024). Providing clarity on these references would improve the reader’s understanding.

- Some networks are indicated with indexed parameters (e.g., the encoder q_{\theta}), while others, such as \psi, are not. Consistent notation would make it clearer which terms refer to networks.

- I am not familiar with BFNs, but I was wondering why are “the series of latent semantics $\{z_t\}_{t=1}^T$ expected to exhibit progressive semantic changes (e.g., age, smile, skin color)” if they encode the parameters of the data distribution at each time-step (line 177)? Given that the entire distribution is refined over time, what drives the latents across time-steps to model intra-distribution features like age or skin color? Is there a formal justification or an explanation to support this claim?

- Could the authors clarify the notation “p(. |-)” in line 252-253, as "-" is not immediately familiar?

- How does the size of the latent representation $z$ compare to that of the parameters $\theta$?

**4 Related Work**
- In describing diffusion models, it may be helpful to avoid saying they are “unsuitable” for representation learning without a clear definition of what is meant by “representation learning.” If the authors refer to representation learning as achieving a continuous, lower-dimensional latent space, this distinction should be made clear. Otherwise, the later point about pre-trained diffusion models being used for other tasks, which is related to representation learning, could appear contradictory.

- How does ParamReL differ from InfoDiffusion in terms of learning latent representations?

**5 Experiments**
- The authors claim to address mixed-type data yet only experiment with discrete and continuous distributions—no mixed-type data is tested, and the only discrete data is binary. To strengthen the contrast with methods like InfoDiffusion, more emphasis on mixed-type data experiments, especially with categorical data, would be helpful.

- In line 330, could the authors define MMD?

---

### Official Review · Reviewer_muVs · 2024-11-02

**Soundness:** 3
**Presentation:** 1
**Contribution:** 2
**Rating:** 3
**Confidence:** 3

**Summary:**

Inspired by recent works which demonstrate that the pretrained (network) parameters in diffusion models encode sensible, semantic information that can be leveraged in downstream tasks (see e.g. Baranchuk et al., 2021), the authors proposed a modification to the recently introduced Bayesian flow networks (BFNs), to infer low-dimensional latent representations encoding similar semantic information.

Like diffusion models, which are trained to progressively denoise some artificially corrupted input data, BFNs learn to progressively modify the parameters of their so-called *output distribution*. Note that, since the BFNs’ dynamics take place in the parameter space of the output distribution, they can readily be used to handle discrete and continuous data. Also note that the dimension of the parameters space and that of the data is the same in BFNs.

The key ideas behind the authors’ proposal are (i) to introduce a *self-encoder*, which maps the time-dependent, progressively-modified parameter of BFNs into a lower dimensional latent representation; and (ii) to modify the output distribution of BFNs to take as input not only its progressively-learned parameter, but also the newly inferred latent representation. To ensure effective learning of these representations, the authors maximize the mutual information between the distribution's parameters and the latent code.

Through a series of experiments, the authors demonstrate the quality of the inferred low-dimensional representations, which outperform those from state-of-the-art diffusion-based approaches.

**References**

Baranchuk et al. [2021]: Label-Efficient Semantic Segmentation with Diffusion Models

**Strengths:**

1. This work represents an elegant alternative to diffusion-based models for representation learning;
2. The authors demonstrate that their methodology can indeed by used to infer latent representations from both continuous and discrete data, and compare the quality of said representations against those of both classical and very recent baselines, which gives credibility to their results (see however the weaknesses below);
3. The authors investigate the quality of the content encoded by their inferred representations with a large set of experiments;
4. This work could motivate further research that aim to increase our understanding of the type of information that is encoded into the BFNs' sequential process.

**Weaknesses:**

Despite its merits, especially their many experiments, I think this work needs significant rewriting before it can be published.

1. The paper contains numerous typos and grammatical mistakes, which make it difficult for readers to understand its content. The paragraphs in lines 71-78 and 181-186 are but two examples. Other paragraphs not only contain such mistakes but are also written in a way that makes them difficult for readers to follow. Unfortunately, most of the experimental section, specially section 5.3, feature such problems. Thus, despite the numerous experiments, the writing style makes it complicated for the reader to go through the experimental section and, consequently, successfully judge the proposed method. I also note that Table 1 is never referenced in the main text.

2. Although the authors compare against recent diffusion-based representation learning algorithms, they do not explain what the differences are between these baselines and their proposal. Why is their proposal interesting, beyond the ability of BFNs to naturally handle discrete and continuous data, as compared to the baselines? More importantly, why does it work better than the baselines? I’d suggest the authors include such discussions either in their related work or the experiment section. Likewise, I’d suggest that they also include answers to questions 1 and 2 below, or at least some content in the same direction.

3. Many of the details and reasoning behind some aspects of the model are left out, or at least I couldn’t find them in the paper. See for example questions 3 and 4 below. Adding such information back into the paper will improve its presentation.

**Questions:**

1. Diffusion-based representation learning algorithms like InfoDiffusion infer a latent representation from data space directly into latent space. Your proposal infers latent representations from parameter space. Both the data space in InfoDiffusion and the parameter space in your case have the same dimension, don’t they?. What would you say is the reason for the better performance of ParamReL, as compared with InfoDiffusion?
2. Opposite to InfoDiffusion, Rombach et al. [2022], whom the authors cite, trains a diffusion model directly on representation space. Has anyone tried to study the quality of the representations learned by Rombach et al. [2022] *from continuous-data* as you do in Table 1? What would you say are the key differences between their representations and yours, *in the continuous-data case*? What makes your approach better?
3. Is the prior distribution over the latent code $p(\mathbf{z}\_t)$ independent of time? In line 171 you write that it "follows a Gaussian distribution", but do the parameters of the Gaussian change with time? If not, why did you decide to constrain the time-dependent posterior with a static prior?
4. Did you include the actual expressions for the inverse of the Bayesian update function $h^{-1}$ somewhere in the paper? I don’t manage to find them.

**References**

- Wang et al. [2023]: InfoDiffusion: Representation Learning Using Information Maximizing Diffusion Models
- Rombach et al. [2022]: High-Resolution Image Synthesis with Latent Diffusion Models

---

### Official Review · Reviewer_W3SV · 2024-11-04

**Soundness:** 2
**Presentation:** 2
**Contribution:** 3
**Rating:** 5
**Confidence:** 2

**Summary:**

The paper treats representation learning and introduces ParamReL, an extension to Bayesian Flow Networks to learn parameter space latents semantics. The motivation is the ability of BFNs to unify representation learning to mixed data modalities. The extension allows learning low dimensional latents by introducing two new components: a self-encoder that learns low-dimensional latent semantics from intermediate parameters and a conditional decoder that generates outputs based on both the latents and parameters. Here these are implemented as a U-Net. These are trained by optimizing a loss capturing the ELBO and a weighted mutual information term between the latent and parameters. ParamReL permits sampling and reverse-sampling which are both described in the paper, and allow reconstruction and interpolation of samples.

The experiments are across multiple datasets and their binarized versions (MNIST, FashionMNIST, CelebA, CIFAR10, Shapes3D). The method's representations are evaluated on downstream classification problems. Additionally, ParamReL is evaluated on reconstruction, interpolation and disentanglement. These results are compared to various baselines. All the objective results presented favor variants of ParamReL over the other baselines.

**Strengths:**

- Novel and well-motivated approach to parameter space representation learning. The unification of modeling of different data modalities should be of high practical value.
- The paper provides a clear description of the fundamentals of the proposed method, clearly illustrating the introduced core components.
- Comprehensive empirical evaluation on both generation and downstream tasks.
- The proposed method performs well on the paper's benchmarks (though there is a large discrepancy between the results reported here and in the baseline methods papers that should be addressed).

**Weaknesses:**

- The paper refers to itself as a method to learn latent semantics of mixed data types (title, motivation, etc). Yet this ability is not experimentally verified, all experiments are on individual data modalities.
- Having to specify a hyperparameter to control the trade-off between the reconstruction quality of the model and the representation quality will be difficult in practice.
- Details beyond the core components would benefit from clarification across the paper. Examples include:
  - The experimental protocol, including how the various metrics such as FID were evaluated.
  - The exact parameterizations for the different experiments. There seems to be some detail in table 2, though how exactly it relates to e.g. table 1 should be clarified.
  - The total compute time for training
- There is no ablation study. In particular there are no experiments without the mutual information score and there is no comparison to the original BFN.
- The linked code repository is empty.

**Questions:**

- Why is the evaluation protocol different for continuous and discrete data, and apart from changing the parameter space, are there other differences between the continuous and discrete models?
- Can you elaborate on the significance of the lambda parameter in tables 1 and 4?
- Looking at table 1, you report significantly higher FID scores for DiffAE than was reported in the cited paper. Can you please explain why this is?
- Can you please elaborate on the following statement for figure 4? “(b), the learned semantics exhibit progressive, time-varying changes”

---

### Meta-Review · Area_Chair_4Xmx · 2024-12-17

**Metareview:**

This paper extends Bayesian Flow Networks (BFNs) to handle mixed-type data. The idea is to introduce another set of lower-dimensional latent variables via encoding the intermediate parameter variables in BFNs, and the hope is that this set of "latent semantics variables" can gradually recover useful representations from the data and their noisy versions.

While the reviewers appreciated the paper's clear presentation of BFNs and some of their approaches, they raised the following issues:

(1) No experiment on mixed-type data despite having "mixed-type data" in the title and motivations.

(2) Comparisons to "Wang et al. [2023]: InfoDiffusion: Representation Learning Using Information Maximizing Diffusion Models".

There is no author rebuttal so the concerns were not addressed. In this situation the submission cannot be accepted in its current form.

**Additional Comments On Reviewer Discussion:**

The authors did not participate in the rebuttal process, so no further information here.

---

### Decision · Program_Chairs · 2025-01-22

Reject